# NerVE: Nonlinear Eigenspectrum Dynamics in LLM Feed-Forward Networks

**Nandan Kumar Jha & Brandon Reagen**
New York University
{nj2049,bjr5}@nyu.edu

## Abstract

We introduce NerVE, a unified eigenspectral framework for understanding how feed-forward networks (FFNs) in large language models (LLMs) organize and regulate information flow in high-dimensional latent space. Despite FFNs dominating the parameter budget, their high-dimensional dynamics remain poorly understood. NerVE addresses this gap through lightweight, memory-efficient tracking of eigenspectrum dynamics via four complementary metrics: Spectral Entropy (dispersion), Participation Ratio (effective dimensionality), Eigenvalue Early Enrichment (top-heaviness), and Jensen-Shannon divergence (distributional shifts). Our *key insight* is that FFN nonlinearities reinject variance across eigenmodes, fundamentally governing latent dimension utilization, and that optimizer geometry strongly modulates the extent of this variance reinjection. We validate NerVE across model scales, and diverse architectural and optimizer configurations, each uniquely shaping FFN dynamics: normalization schemes controlling variance flow; FFN weight geometries constraining latent space; positional encoding and activation functions regulating information flow; and optimizer choices redistributing effective capacity across depth. Across these settings, NerVE consistently recovers stable spectral signatures that correlate with model's generalization ability and respond predictably to design choices, generalizing beyond transformer to MLP-Mixer architectures, providing actionable insights for architectural and optimizer choices beyond trial-and-error.

## 1 Introduction

Large Language Models (LLMs) have demonstrated remarkable capabilities across a wide range of natural language tasks, driven in part by advances in transformer-based architectures. While much emphasis has been devoted to understanding attention mechanisms and token-wise interactions, the role of feed-forward networks (FFNs), particularly their nonlinear components, remains underexplored, despite FFNs dominating both the parameter budget and computational footprint of transformer-based models (Geva et al., 2021; de Vries, 2023).

Despite their apparent simplicity, FFNs perform high-dimensional *nonlinear* transformations that regulate information flow by reorganizing, compressing, and propagating the information extracted by attention modules across layers. Understanding how these transformations evolve and interact with architectural design choices remains a fundamental open question.

One challenge in interpreting FFNs is the absence of systematic and efficient tools for characterizing how latent representations are structured and transformed by nonlinear activations. FFN transformations unfold in a high-dimensional latent space which is far less accessible for direct visualization and probing compared to multi-head attention. Prior work, Kobayashi et al. (2024) used attention maps to study the input-contextualization effect of FFNs, and Balestriero et al. (2024) characterized FFN geometry through piecewise-affine spline partitions. Neither lens reveals how nonlinearity redistributes variance, nor captures the rich spectral structure inherent in these transformations.

To this end, we introduce NerVE, a unified, online, and memory-efficient framework for analyzing FFN latent geometry through the eigenspectrum analysis. NerVE summarizes pre- and post-activation spectra using four scale-invariant, distribution-aware metrics: spectral entropy (dispersion vs uniformity), participation ratio (effective latent dimensionality), eigenvalue early enrichment (top-heaviness), and Jensen-Shannon divergence (distributional shift).

Figure 1: NerVE quantifies nonlinear eigenspectrum dynamics in FFNs of GPT-2. FFN nonlinearity (GELU) regulates information flow by reinjecting variance, reactivating under-utilized directions (post-activation SE↑ and PR↑), and flattening the eigenspectrum, less top-heavy (post-activation EEE↓). The JS heatmap shows a depth-localized transition band where redistribution is strongest.

From a methodological standpoint, these metrics span a broad theoretical range and expose the complementary facets of the eigenspectrum that any single scalar would obscure, thereby enabling continuous tracking of latent geometric dynamics. Spectral Entropy (SE) captures the uniformity of variance distribution (De Domenico & Biamonte, 2016), and Participation Ratio (PR) reflects the geometric notion of effective dimensionality, indicating how many directions meaningfully contribute to total variance (Gao et al., 2017). Unlike SE and PR, Eigenvalue Early Enrichment, which quantify the top-heaviness, can distinguish the eigenspectrum utilizing different fractions of the latent space (Marbut et al., 2023). Finally, the Jensen-Shannon (JS) divergence provides an information-theoretic distance measure between two eigenspectra (Lin, 1991), quantifying distributional shifts of variance.

We apply this framework across a diverse range of architectural settings, including LayerNorm placements: PreLN, PostLN, and MixLN (Li et al., 2025); normalization-free variants (Jha & Reagen, 2024); FFN weight-geometry constraints (Miyato et al., 2018; Salimans & Kingma, 2016), and hyperspherical constraints (Liu et al., 2017); positional encoding scheme (Su et al., 2024); and the optimizer choices, including Adam (Kingma, 2015; Loshchilov & Hutter, 2019), Muon (Jordan et al., 2024), Dion (Ahn et al., 2025), Adafactor (Shazeer & Stern, 2018), and SGD; see Table 1.

Across settings, *a clear pattern emerges:* FFN nonlinearities do not merely rescale the activations, they actively reinject the variance across eigenmodes and reawakens the inactive directions in high-dimensional latent space. As shown in Figure 1, the post-activation spectra in GPT-2 consistently show increases in SE and PR, and decreases in EEE, while JS heatmaps reveal depth-localized transition bands where redistribution is strongest. This highlight the active role of FFN nonlinearities in regulating information flow and latent geometry that downstream layers further exploit.

**Contributions:** Our contributions can be summarized as follows:

1. **Conceptual.** We demonstrate that FFN nonlinearities do not simply rescale activations but actively reorganize eigenspectra, reinjecting variance into under-utilized directions. Moreover, optimizer geometry modulates the extent of variance reinjection, altering the role of FFN nonlinearity from repair (recovering spectral collapse) to refinement (stabilizing a well-conditioned spectrum).
2. **Framework.** We introduce NerVE, a lightweight and memory-efficient methodology for online tracking of FFN eigenspectrum dynamics, using four distribution-aware, scale-invariant metrics.
3. **Diagnostic.** We show that architectural (normalization layers, activation functions, gating, weight geometry, positional encodings), and optimizer (AdamW, Muon, Dion, Adafactor, SGD) choices imprint distinct spectral signatures in FFNs, which can be used for diagnosing the model behaviors.
4. **Empirical.** We validate NerVE on GPT-2 and LLaMA models (71M to 1.3B) trained from scratch on CodeParrot, OpenWebText, FineWeb, and C4 datasets; extend to the non-transformer MLP-Mixer (B/16) on CIFAR-100, confirming cross-architecture generality; and perform extensive robustness studies across normalization variants, optimizer family, and token positions.

Table 1: Summary of key NerVE findings per experimental axis.

| Experimental Axis | Key Findings |
| --- | --- |
| Activation (GELU vs ReLU) | Similar trend, distinct dynamics; GELU explores broader subspace (§3.1) |
| Norm-free models | GELU exhibits spectral inertia; ReLU compensates LayerNorms (§3.2) |
| FFN weight geometry | Spectral dynamics matter; performance tracks sustained flattening (§3.3) |
| Norm placement (Pre/Post/Mix) | PreLN: best return-on-width; PostLN: diminishing spectral returns (§3.4) |
| Positional encoding | RoPE prevents mid-to-deep spectral collapse (depth utilization ↑) (§3.6) |
| Optimizer (AdamW vs Muon) | Repair or refinement; performance follows mid-layer capacity trends (§3.7) |
| Token positions | Position-uniform early on; depth shift capacity to later-tokens (Appendix J) |
| Non-transformer (MLP-Mixer) | Core findings generalize beyond transformer architecture (Appendix I) |
| LayerNorm vs RMSNorm | Robust; placement and FFN activation choices matter more (Appendix K) |

## 2 NerVE: A Principled Framework for Eigenspectrum Analysis

**Notations.** Let $L$ be the number of layers, $d$ the embedding dimension, $D$ the FFN hidden dimension, $B$ the batch size, $S$ the context length, and $\Sigma$ the FFN (pre/post-activation) covariance matrix.

### 2.1 Formulation of Eigenspectrum-Based Framework

To understand how information is structured and propagated through FFN latent space, we analyze: (1) variance distribution and its impact on effective dimensionality; (2) how nonlinearity within a layer reshapes this distribution; (3) how these patterns evolve across layers and training. The NerVE framework (Figure 10 in Appendix A.1) consists of four main components: i) activation collection, (ii) covariance matrix computation, (iii) eigendecomposition, and (iv) spectral metrics calculation.

**Activation collection.** For a FFN with (non-gating) architecture $\text{FFN}(x) = W_{\text{down}}\sigma(W_{\text{up}}x + b_1) + b_2$, where $\sigma$ is the activation function (*e.g.*, ReLU, GELU), we collect $\text{PreAct}(X) = W_{\text{up}}x + b_1$ and $\text{PostAct}(X) = \sigma(W_{\text{up}}x + b_1)$, the output of the up projection, and input to the down projection (after activation function), respectively. For activation with gating mechanisms (e.g., SwiGLU in LLaMA), the architecture becomes $\text{FFN}(x) = W_{\text{down}}(\sigma(W_{\text{gate}}x) \odot (W_{\text{up}}x))$, where $\odot$ denotes element-wise multiplication, and we collect $\text{PreAct}(X) = W_{\text{gate}}x$ and $\text{PostAct}(X) = \sigma(W_{\text{gate}}x) \odot (W_{\text{up}}x)$.

**Covariance matrix computation**. At the logging step $t$, for each layer $l$, we collect full activation matrices $\text{PreAct}(X^{(l,t)}) \in \mathbb{R}^{N \times D}$ and $\text{PostAct}(X^{(l,t)}) \in \mathbb{R}^{N \times D}$, where $N = B \times S$ is the total number of tokens in the batch. These tensors, originally shaped $[B, S, D]$, are flattened to $[B \times S, D]$, intentionally discarding sequence order. This allows us to compute an unbiased covariance matrix for all tokens in the batch, treating each token as an independent sample in FFN latent space.

Computing covariance with all $N$ tokens in a batch, with no sub-sampling, ensures *exact* second-order statistics rather than their statistical approximations; thus, spectral analysis captures *true* statistical properties of distribution. For each set of activations, we compute covariance matrix as follows:

$$\Sigma = \frac{(X - \mu)^T(X - \mu)}{N - 1} \in \mathbb{R}^{D \times D,} \text{ where } X \in \mathbb{R}^{N \times D} \text{ are activations and } \mu = \frac{1}{N}\sum_{i=1}^{N}X_i \quad (1)$$

This yields two covariance matrices per FFN layer: $\Sigma_{\text{PreAct}}^{(l,t)}(X)$ and $\Sigma_{\text{PostAct}}^{(l,t)}(X)$.

**Eigendecomposition** For each covariance matrix, we perform eigendecomposition ($\Sigma v = \lambda v$), and sorted the eigenvalues in descending order: $\lambda_1 \geq \lambda_2 \geq \ldots \geq \lambda_D \geq 0$. We define $\Lambda = \sum_{i=1}^{D}\lambda_i$, the total variance, and normalized eigenvalues to create a probability distribution as $\hat{\lambda}_i = \lambda_i/\Lambda$.

**Spectral metrics computation** Next, we compute four scalar metrics from the eigenspectrum of $\Sigma_{\text{PreAct}}^{(l,t)}(X)$ and $\Sigma_{\text{PostAct}}^{(l,t)}(X)$, which quantify distinct aspects of the eigenspectral dynamics: Spectral Entropy, Participation Ratio, Eigenvalue Early Enrichment, and the Jensen-Shannon divergence.

### 2.2 Eigenspectrum Metrics for Analyzing High-Dimensional Latent Space

**Spectral Entropy (SE)** Spectral Entropy quantifies the uniformity of eigenvalue distribution in high-dimensional latent spaces. Formally, it is the Shannon entropy of the normalized eigenvalue distribution derived from a layer's covariance matrix: $\text{SE} = -\sum_{i=1}^{D}\hat{\lambda}_i \log \hat{\lambda}_i$.

Mathematically, spectral entropy is equivalent to the von Neumann entropy (vNE) in quantum information theory, which quantifies the degree of quantum entanglement or *mixedness* of a quantum state (De Domenico & Biamonte, 2016). In quantum mechanics, vNE is defined as: $S_{\text{vNE}}(\rho) = -\text{Tr}(\rho \ln \rho)$, where $\rho$ denotes a density matrix, a positive semidefinite operator with unit trace that encapsulates the probabilistic nature of quantum states (Nikitin et al., 2024; Huang et al., 2023).

For FFN, an analogous density matrix is created by normalizing the covariance matrix by its trace: $\rho_{\text{FFN}} = \frac{\Sigma}{\text{Tr}(\Sigma)}$, where $\text{Tr}(\Sigma) = \Lambda$. Applying this to $\rho_{\text{FFN}}$, SE becomes the Shannon (or von Neumann) entropy of the normalized eigenvalue distribution $\text{SE} = -\sum_{i=1}^{D}\hat{\lambda}_i \log \hat{\lambda}_i$.

Thus, when the eigenspectrum exhibits significant anisotropy (e.g., $\lambda_1 \gg \lambda_2, \ldots, \lambda_D$), SE approaches zero, indicating a collapsed or low-rank representation. Conversely, when eigenspectrum approach uniformity ($\lambda_i \approx \lambda_j : \forall i, j$), SE approaches its theoretical maximum, $\ln(D)$.

**Participation Ratio (PR)**. It measures *effective dimensionality* of an eigenspectrum (Hu & Sompolinsky, 2022) and quantifies how many dimensions significantly hold variance. Formally,

$$\text{PR} = \frac{\left(\sum_{i=1}^{D} \lambda_i\right)^2}{\sum_{i=1}^{D} \lambda_i^2} = \frac{\Lambda^2}{\sum_i \lambda_i^2} = \frac{1}{\sum_i \hat{\lambda_i}^2}; \quad \text{where} \quad 1 \leq \text{PR} \leq D \tag{2}$$

PR values close to 1 indicate maximal anisotropy (i.e., variance concentrated in a single direction), while a value near $D$ indicates uniform variance across all dimensions. While SE depends on the entire distribution shape (including small eigenvalues) and measures the uniformity of distribution, PR focuses on how many directions are active and meaningfully contributing to the total variance.

**Early Eigenvalue Enrichment (EEE).** It quantifies the *top-heaviness* of an eigenspectrum by tracking how rapidly the leading principal directions accumulate variance. Specifically, it captures how *front-loaded* the variance is among the top eigenvalues, by assessing how quickly the cumulative sum surpasses that of a uniform spectrum (Marbut et al., 2023).

Formally, the proportion of variance explained by the top $k$ principal directions is defined by the normalized cumulative sum at index $k$ as $\widetilde{S}_k = \frac{1}{\Lambda} \sum_{i=1}^{k} \lambda_i$, and for comparison, the ideal uniform reference grows linearly as $\frac{k}{D}$ (see Figure 2). The EEE score is then the average vertical distance between the empirical cumulative curve and this ideal line, normalized by the maximal possible value:

$$\text{EEE} = \frac{1}{\frac{1}{2}D} \sum_{k=1}^{D} \left(\widetilde{S}_k - \frac{k}{D}\right) = 2 \times \sum_{k=1}^{D} \left(\frac{\sum_{i=1}^{k} \lambda_i}{\sum_{i=1}^{D} \lambda_i} - \frac{k}{D}\right) \times \frac{1}{D}. \tag{3}$$

EEE $\approx 1$ indicates that most of the variance is concentrated in the top few directions, forming a steep eigenvalue spectrum; conversely, EEE $\approx 0$ corresponds to a nearly uniform spectrum.

We analyze the cumulative variance distribution of various eigenspectra over a 768-dimensional latent space, using the EEE metric in Figure 2. The resulting curves shows a spectrum of dimensional utilization, ranging from extreme anisotropy to fully uniform variance. The One dimension spectrum exhibits an EEE of 1.00, where nearly all variance is concentrated in a single dominant principal component, indicative of a highly *degenerate* latent representation. As more dimensions begin to carry variance (from 10% to 99%), the EEE value decreases from 0.94 to 0.44, suggesting a gradual transition toward more distributed representations. Notably, EEE's nonlinear scaling with dimension count highlights its sensitivity to early eigenvalue dominance, making it a valuable diagnostic for understanding how architectural choices and training dynamics shape the effective dimensionality of latent spaces.

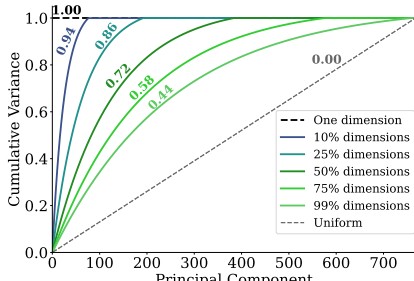

Figure 2: Cumulative variance distribution across a 768-dimensional latent space. Higher values (shown on curves) indicate top-heavy concentration in a few dominant directions, while lower values reflect a more uniform distribution.

**Jensen-Shannon Divergence (JS)** Unlike the previous metrics, which describe a single eigenspectra in isolation, JS provides a principled measure of dissimilarity between two eigenspectrum within a layer. Specifically, it quantify the *extent of distributional shifts* from the *pre* to *post* eigenspectrum caused by FFN nonlinearity. For a normalized eigenvalue distributions $P_{\text{pre}} = \{\hat{\lambda}_i^{\text{pre}}\}_{i=1}^{D}$ and $P_{\text{post}} = \{\hat{\lambda}_i^{\text{post}}\}_{i=1}^{D}$, where $\hat{\lambda}_i = \lambda_i / \sum_{j=1}^{D} \lambda_j$, the JS is defined as (see Amari (2016, Chapter 4.6.3)):

$$\text{JS}(P_{\text{pre}} \parallel P_{\text{post}}) = \frac{1}{2} D_{\text{KL}}(P_{\text{pre}} \parallel M) + \frac{1}{2} D_{\text{KL}}(P_{\text{post}} \parallel M) \tag{4}$$

where $M = \frac{P_{\text{pre}} + P_{\text{post}}}{2}$ is the midpoint distribution and $D_{\text{KL}}$ is Kullback-Leibler divergence:

$$D_{\text{KL}}(P \parallel Q) = \sum_{i=1}^{D} \hat{\lambda}_i^P \log\left(\frac{\hat{\lambda}_i^P}{\hat{\lambda}_i^Q}\right) \tag{5}$$

For numerical stability, we compute JS for FFN as follows:

$$\text{JS}(P_{\text{pre}} \parallel P_{\text{post}}) = \frac{1}{2} \sum_{i=1}^{D} \hat{\lambda}_i^{\text{pre}} \log\left(\frac{2\hat{\lambda}_i^{\text{pre}}}{\hat{\lambda}_i^{\text{pre}} + \hat{\lambda}_i^{\text{post}}}\right) + \frac{1}{2} \sum_{i=1}^{D} \hat{\lambda}_i^{\text{post}} \log\left(\frac{2\hat{\lambda}_i^{\text{post}}}{\hat{\lambda}_i^{\text{pre}} + \hat{\lambda}_i^{\text{post}}}\right) \tag{6}$$

## 3 EXPERIMENTAL RESULTS

**Models and datasets** We evaluate the FFN eigenspectrum of two model families: GPT-2 and LLaMA-style architectures. For GPT-2, we train a 125M parameter model on 2.1B tokens from the CodeParrot dataset, which is created from 20M GitHub Python files and preprocessed using HuggingFace tokenizer of vocabulary size 50K. For LLaMA-style models, we train in-house variants with 71M and 130M parameters on the C4 dataset (Raffel et al., 2020), tokenized using the T5-base tokenizer with a 32K vocabulary. These LLaMA variants follow the architectural specifications (depth, embedding dimensions, FFN width, positional encoding, and SwiGLU activation) from Li et al. (2025), which adopts downscaling methodology of Lialin et al. (2024); Zhao et al. (2024). For experiments with RoPE, we train GPT-2 on OpenWebText dataset, following the architectural settings and training recipe from Loshchilov et al. (2025). To study optimizer-dependent (AdamW, Muon, Dion) dynamics, we train GPT-2 350M and 160M variants on FineWeb (Penedo et al., 2024) dataset.

**Training setup** All experiments are conducted on NVIDIA RTX 3090 GPUs (24 GB). GPT-2 models are trained for 41K steps with context length 128 on the CodeParrot dataset. For RoPE experiments, GPT-2 is trained on 26B tokens from OpenWebText using 4 GPUs with context length 512. LLaMA-71M is trained on 1.1B tokens for 10K steps, while LLaMA-130M, LLaMA-250M, and LLaMA-1.3B variants are trained on 2.2B tokens for 20K steps. All LLaMA models use a context length of 256. For optimizer-specific eigenspectrum analysis, we train GPT-2 models with 512 and 1024 context lengths, following the hyperparameter settings from Ahn et al. (2025).

### 3.1 FFN NONLINEARITY REINJECT VARIANCE AND FLATTEN THE EIGENSPECTRUM

**Variance is reinjected, not merely rescaled** Figure 1 contrasts pre- and post-activation spectral dynamics and highlights the role of nonlinearity withing FFN. The PreAct eigenspectrum is highly top-heavy as most variance is concentrated in a few leading directions. This is reflected by lower SE and PR, indicating a lower utilization of the latent space. Once the nonlinearity is activated, both SE and PR jump upward across training, suggesting that the nonlinearity redistributes variance across more dimensions. In effect, the nonlinearity *reawakens* previously inactive directions, injecting new degrees of freedom into the latent space. This variance reinjection promotes features disentanglement which facilitate more effective downstream processing in subsequent layers.

**Flattening and reshaping the eigenspectrum** The variance redistribution has a noticeable impact on the spectrum shape. The EEE values, which quantifies how sharply leading eigenvalues dominate, drops consistently for post-activation, re-affirming that the spectrum is being flattened. Instead of concentrating variance in a small number of dominant modes, the post-activation spectrum spreads variance more evenly. Moreover, the JS heatmaps shows a distributional shift: post-activation eigenspectra are not merely scaled versions of pre-activation ones but are effectively reordered.

**GELU vs. ReLU: Similar Trajectory, Distinct Dynamics** GELU (Figure 1) and ReLU (Figure 3) follow the same qualitative trajectory—variance reinjection ($\uparrow$SE, $\uparrow$PR), spectral flattening ($\downarrow$EEE), and distributional reordering ($\uparrow$JS)—but differ in pace and extent. While ReLU stabilize SE and PR earlier, suggesting a faster reinjection of variance, GELU progresses more gradually yet ultimately pushes $PR_{post}$ to higher values. This indicates that smoother nonlinearity in GELU enables a broader subspace exploration, which correlates with GELU's lower perplexity (Table 2).

These eigen-metrics highlight the functional role of nonlinearity: (1) improving the directional usage of FFN latent space (SE $\uparrow$, PR $\uparrow$), reflecting an increased participation of multiple latent directions for encoding information; and (2) reducing the dominance of a few principal direction (EEE $\downarrow$). Thus, NerVE provides geometric underpinning of the nonlinear expressivity in transformer models.

### 3.2 COMPENSATORY ROLE OF FFN NONLINEARITY IN THE ABSENCE OF LAYERNORMS

Removing LayerNorm from transformer architectures eliminates their layerwise re-centering and variance normalization, shifting the burden of statistical regularization entirely onto the attention and FFN sub-blocks. This motivates a central question: *Can FFN activation functions compensate for the absence of normalization, and if so, to what extent and through what mechanisms?* Our findings reveal that, unlike GELU, ReLU-family activations actively compensate the absence for removal of LayerNorms by regulating the FFN latent space variance.

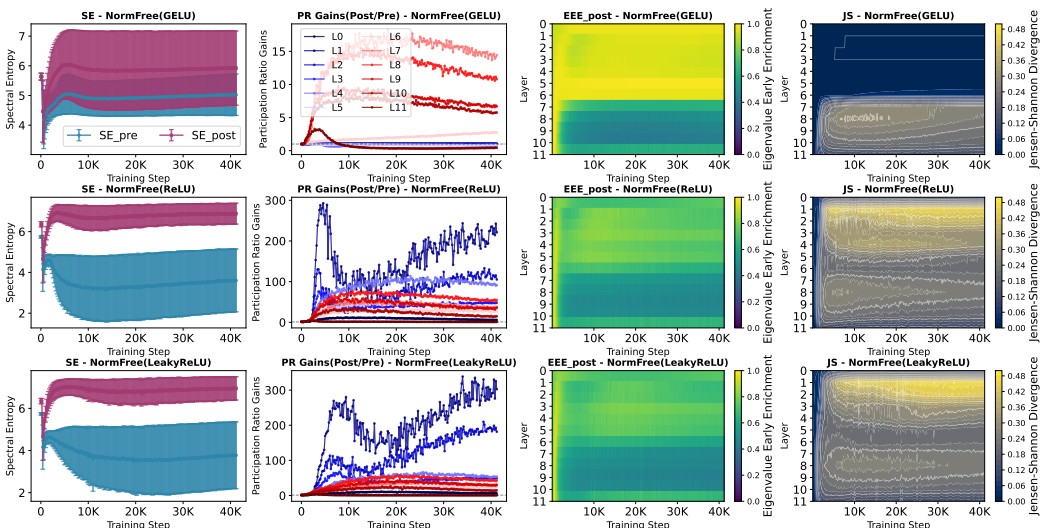

Figure 3: Eigenspectrum dynamics illustrate *how FFN nonlinearities regulate information flow and reshape the eigenspectrum* during training for GPT-2 (ReLU) on CodeParrot. Pre- and post-activation dynamics are shown for SE, PR, and EEE, highlighting how nonlinearities reinject variance and alter spectral structure. JS heatmaps (rightmost) capture the layer-wise distributional shift induced by nonlinearity. In-panel titles report Pearson correlations ($r$) between each metric and evaluation loss.

Figure 4: Eigenspectrum dynamics for norm-free GPT-2 (125M) models with GELU (top), ReLU (middle), and learnable-slope Leaky ReLU (bottom). Columns show layer-averaged SE (pre vs. post), PR gain (post to pre), post-activation EEE (yellow regions indicate top-heavy distribution), and JS (yellow regions highlight strong redistribution) across layers and training steps. Norm-free GELU exhibits spectral inertia in layers 0 to 5 (EEE $\rightarrow$ 1, JS $\rightarrow$ 0); whereas, *ReLU and Leaky ReLU aggressively reinject variance* (PR gain $>$ **200**$\times$) and flattening the spectrum (EEE $<$ 0.3).

**Spectral inertia in normalization-free GELU models** Normalization-free GELU model exhibits spectral inertia in early layers, characterized by EEE$_\text{post}$ $\approx$ 1 and JS $\approx$ 0 (see Figure 4). This indicates that the nonlinearity in early FFNs fails to reinject variance into the latent space, leaving the eigenspectrum heavily front-loaded. Consequently, variance remains confined to a few dominant subspaces, and there is a significant overlap between SE$_\text{pre}$ and SE$_\text{post}$. Thus, nonlinearity in early FFNs does not activate new directions, and information continues to flow through a narrow subspace in subsequent layers. This *spectral bottleneck* reflects a downstream consequence of entropic overload, a critical failure mode observed in normalization-free LLMs (Jha & Reagen, 2024), where a disproportionate number of attention heads in the early layers stuck in higher-entropy states throughout training, squandering the representation diversity of multi-head attention mechanism, and degrades the performance (higher perplexity, see Table 2).

**Early FFNs overcompensate to break spectral inertia in normalization-free ReLU models** In contrast with GELU, ReLU and learnable-slope Leaky ReLU variant exhibit strong compensatory behavior when LayerNorms are removed. Specifically, in the first two FFN layers, the post-to-pre Participation Ratio (PR) gain surges by $\approx$ 20$\times$ to 300$\times$ (blue curves, Figure 4), indicating an abrupt reinjection of variance into previously underutilized latent directions. Consequently, the post-activation EEE$_\text{post}$ remains consistently low ($\approx$ 0.3-0.5) across layers, indicating that the spectrum becomes flatter and more isotropic, rather than top-heavy. This redistribution is further corroborated by non-overlapping SEpre and SEpost spectrum, and by JS peaks $\approx$0.48 in the early-layer contour maps, confirming the crucial role of nonlinearity in reshaping eigenspectrum in early FFNs

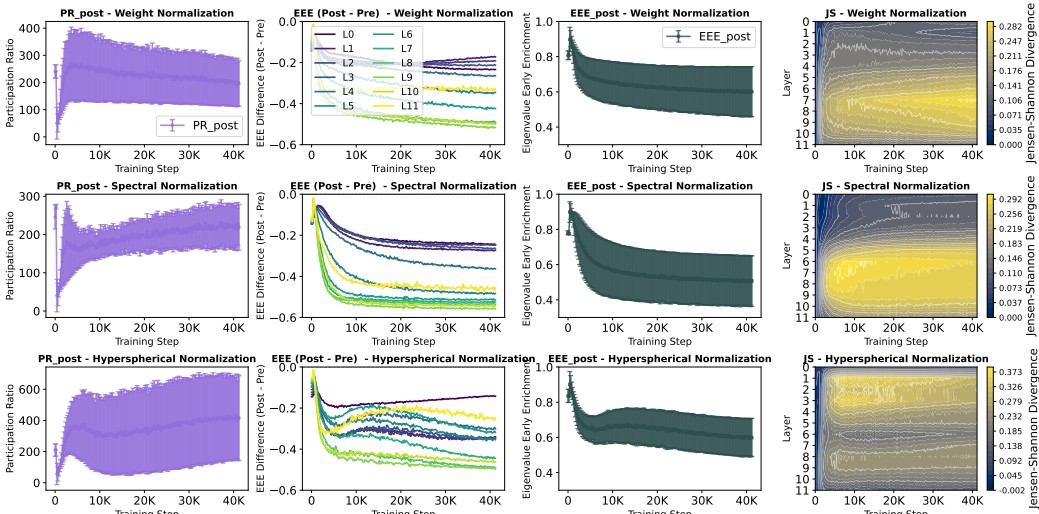

Figure 5: Impact of FFN (parametric) normalization in norm-free GPT-2 with learnable-slope leaky ReLU. Eigenspectrum dynamics are quantified by latent capacity (PR_post), spectral regularization and flattening ($\Delta$EEE and EEE_post), and distributional shift (JS). Top to bottom: Weight, Spectral, and Hyperspherical Normalization. Each method exhibits distinct JS localization and spectral patterns, showing different influences on FFN internal dynamics.

This *aggressive variance injections* demonstrate that FFN nonlinearity can partially assume the statistical regularization role of LayerNorm, widening the latent manifold and mitigating spectral bottlenecks. In terms of predictive performance, both ReLU variants reduce the perplexity gap to the LayerNorm baseline by $\approx$50% (refer to Table 2).

Table 2: Evaluation perplexity (PPL $\downarrow$) comparison across GPT-2 baseline models (GELU and ReLU), norm-free models (GELU, ReLU, learnable-slope Leaky ReLU). Parametric normalization (Weight, Spectral, Hyperspherical) are applied to FFNs of norm-free learnable-slope Leaky ReLU models. All models trained on 2.1B tokens from CodeParrot dataset.

| | Baseline Models | | Norm-free Models | | | Norm-free w/ FFN-Norm | | |
|---|---|---|---|---|---|---|---|---|
| | GELU | ReLU | GELU | ReLU | Leaky ReLU | WNorm | SNorm | HNorm |
| PPL | 2.714 | 2.774 | 3.223 | 2.988 | 3.081 | 3.041 | 3.000 | 3.122 |

## 3.3 FEED-FORWARD NETWORKS WEIGHT GEOMETRY AND EIGENSPECTRUM DYNAMICS

Previously, we have seen that how ReLU variants improve the redistribution of top-heavy eigenvalues in the early layers of normalization-free LLMs. We now analyze how parametric normalization applied to their FFNs further influence eigenspectrum dynamics. Figure 5 shows the effects of weight, spectral, and hyperspherical normalization applied to FFNs.

**Parametric normalization alters the localization of distributional shifts across layers** Despite being applied only to FFN linear layers, each parametric normalization technique induces distinct learning dynamics, as demonstrated by the layerwise JS divergence in Figure 5 (rightmost column). Specifically, SNorm exhibits highly localized distributional shifts in the mid-to-deeper layers that emerge very early in training. In contrast, WNorm induces distributional shifts in a smaller subset of mid layers that appear very late in training. Meanwhile, HNorm triggers strong shifts in the early layers at the very-beginning of training, which gradually diminish as training progresses.

**Spectral normalization achieves superior performance through smooth and sustained spectral flattening** By constraining the spectral norm of each FFN weight matrix, SNorm induces early and consistent spectral flattening, reflected in uniformly negative $\Delta$EEE (Post-Pre) values in Figure 5, especially in deeper layers. This yields the lowest EEE_post ($\approx$-0.45) among all parametric normalization methods, indicating balanced variance distribution across a moderate number of directions (PR_post $\approx$ 200) and improved latent space utilization. In contrast, WNorm shows delayed and highly localized flattening in a few mid layers, while HNorm induces early flattening in shallow layers that vanishes as training progresses.

**Hyperspherical normalization underperforms due to early overshooting in eigenspectrum**
HNorm projects weight vectors onto a unit hypersphere (Loshchilov et al., 2025; Lee et al., 2025a; Wang & Isola, 2020), which rapidly expands latent capacity, indicated by a sharp increase in PR_post (exceeding 600). However, this expansion cause an early-overshooting in $\Delta$EEE dynamics, and EEE_post values remain high across depth, indicating the persistent dominance of a few principal directions. Moreover, the JS divergence patterns also reflect the inefficient use of model's depth. Hence, the combination of early overshooting, lack of spectral control, and depthwise redundancy leads to HNorm's degraded perplexity. While large latent capacity can be beneficial, it must be paired with sustained flattening mechanisms to prevent top-heavy eigenspectrum from re-emerging.

### 3.4 IMPACT OF LAYERNORM POSITIONING ON THE FFN LATENT SPACE DIMENSIONALITY

**PreLN turns width into usable dimensions while PostLN shows diminishing returns at higher width.** Across the FFN-width sweep ($D$=1d–8d), the normalized PR, which reflects the effective utilization of available latent space, for PreLN is highest and remains nearly flat as $D$ increases. Figure 6 shows the layer-consistent behavior for PreLN, highlighting the conversion of added width into usable dimensions. Thus, PreLN offers the best return-on-width.

The width utilization is lowest for PostLN and *decreases with $D$*, revealing growing spectral concentration—added capacity is concentrated into fewer dominant directions instead of broadening the effective dimensionality. MixLN is intermediate with medians between PreLN and PostLN and wider layer-to-layer spread, implying a less stable inductive bias across depth. Thus, LayerNorm placement governs how width is spent. Refer to Table 7 (Appendix D) for raw PR_post values, and Figure 11 (Appendix C.1) for a detailed discussion on spectral signatures.

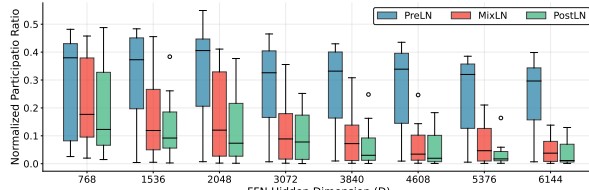

Figure 6: LayerNorm positioning and FFN width sweep: Post-activation participation ratio is normalized by $D$ for PreLN, MixLN, and PostLN configurations. PreLN sustains the highest and most stable utilization of FFN width across the sweep, PostLN incurs diminishing return at higher FFN width, MixLN lies in between but with greater layer-to-layer variability.

### 3.5 EIGENSPECTRAL SIGNATURES PREDICT GENERALIZATION

**Practical utility of NerVE as online monitoring tool and architectural selection proxies.** First, to assess whether NerVE metrics serve as online training diagnostics, we correlate SE and PR with validation loss across training checkpoints for each FFN width variants (Table 3, left). Pre-activation correlations exceed $|r| \geq 0.97$ at every width, indicating that spectral metrics track generalization throughout training, and they can be used as forward-pass-only diagnostic. Notably, the post-activation PR correlation strengthens from $|r| = 0.85$ at $D = 1d$ to $|r| \geq 0.93$ at D $\geq$ 2d, *suggesting that a modest FFN width is required for producing generalization-predictive spectral signatures.*

Table 3: Correlation between eigenspectrum metrics (SE, PR) and generalization. Left (within-run): Pearson $r$ between each metric and validation loss over checkpoints at each FFN width ($D$=1d–8d). Right (cross-config): Pearson $r$ between final metric values and perplexity across the eight width configurations for each architecture and activation. Higher SE and PR consistently implies lower loss.

| Metric | FFN Width Configuration (GPT-2 GELU) | | | | | | | | GPT-2 | | | | NormFree GPT-2 | | |
|---|---|---|---|---|---|---|---|---|---|---|---|---|---|---|---|
| | D=1d | D=2d | D=3d | D=4d | D=5d | D=6d | D=7d | D=8d | GELU | ReLU | GeGLU | SwiGLU | GELU | ReLU | LReLU |
| SE_pre | -0.98 | -0.98 | **-0.99** | **-0.99** | **-0.99** | **-0.99** | **-0.99** | **-0.99** | **-0.99** | -0.98 | -0.95 | -0.97 | -0.82 | 0.03 | 0.03 |
| SE_post | -0.84 | -0.84 | -0.86 | -0.87 | -0.87 | -0.87 | -0.87 | -0.87 | **-1.00** | **-1.00** | -0.57 | -0.85 | -0.92 | **-0.99** | **-1.00** |
| PR_pre | -0.97 | -0.98 | -0.98 | **-0.99** | -0.98 | -0.97 | -0.98 | -0.97 | **-0.99** | -0.98 | -0.97 | -0.97 | -0.93 | -0.55 | -0.60 |
| PR_post | -0.85 | -0.93 | -0.94 | -0.94 | -0.95 | -0.95 | -0.93 | -0.93 | **-1.00** | -0.97 | -0.94 | -0.89 | **-0.99** | -0.94 | **-0.99** |

Second, for cross-configuration ranking, we correlate the final values of eigen-metrics against final perplexity across the eight width configurations, for each architecture and activation variants (Table 3, right). Correlations remain strong across for across the configurations ($|r| \geq 0.85$), with one notable exception—normalization-free ReLU and LeakyReLU, where pre-activation correlations weaken sharply while post-activation correlations strengthen. This inversion directly reflects how *FFN nonlinearity overcompensate to break spectral inertial* identified in Section 3.2, making the post-activation spectrum the more informative diagnostic in this regime. Thus, short preliminary runs with NerVE metrics can rank architectural configurations without training each to convergence.

## 3.6 LAYERWISE DYNAMICS FOR POSITIONAL ENCODING: ROPE VS NOPE

**RoPE prevents mid-to-deep spectral collapse, improving depth utilization**

Figure 7 demonstrate that the NoPE's PR declines in the middle and deeper layers, indicating that representations collapse into a narrow subspace and *squander* model depth. RoPE, on the other hand, sustains higher PR across the mid-to-deeper layers, improving the depth utilization. This effect aligns with recent evidence that intermediate layers are disproportionately important (de Llano et al., 2026; Lad et al., 2025; Ikeda et al., 2025; Skean et al., 2025); which collapse under NoPE but remain effective under RoPE. These improved spectral utilization in RoPE helps achieves lower evaluation perplexity (15.20 vs 16.78) than NoPE. Figure 13 in Appendix C.3 shows the spectral entropy heatmaps reaffirming that RoPE prevents the mid-to-deep spectral collapse characteristic of NoPE.

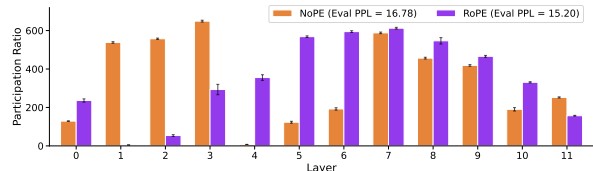

Figure 7: Layerwise participation ratio (PR) comparison (RoPE vs NoPE) in GPT-2 models trained from scratch on 26B token form openwebtext dataset with 512 context length for 100K steps. RoPE sustains higher PR in the middle and deeper layers, indicating better utilization of latent space and network's depth.

## 3.7 OPTIMIZER-DEPENDENT ROLE OF FFN NONLINEARITY: REPAIR VS REFINEMENT

To understand the optimizer-dependent role of FFN nonlinearity, we examine eigenspectrum dynamics under three LLM optimizers; AdamW, Muon, and Dion; and summarize the observations below.

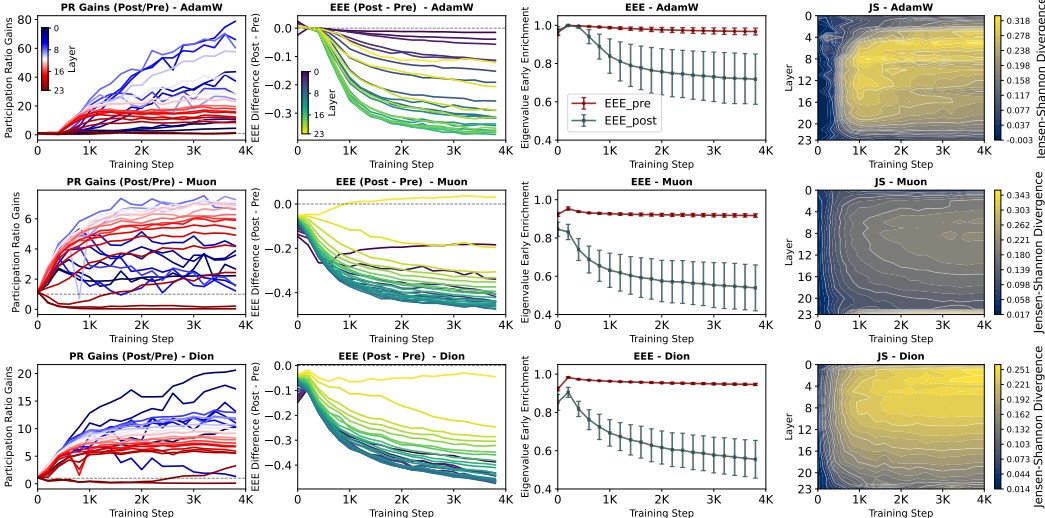

Figure 8: Optimizer-dependent FFN eigenspectrum dynamics in GPT2-350M. Rows show AdamW (top), Muon (middle), and Dion (bottom). AdamW has large early PR gains and high JS with relatively high EEE_post, indicating optimizer-induced pre-activation collapse followed by aggressive but incomplete nonlinear repair. Muon shows the smallest PR gains, lowest JS, and lowest EEE_post, and flatter post-spectra. Dion is intermediate and falls between these two regimes, improving over AdamW but not matching Muon's pre-/post-spectral behavior. The perplexity ordering (Muon > Dion > AdamW) aligns with post-activation spectral flatness.

**Muon minimizes nonlinearity burden by preserving activation-compatible eigenspectrum** Figure 8 shows that Muon maintains uniformly small PR(Post/Pre) gains and consistently low JS divergence throughout training. This combination indicates that Muon keeps the pre-activation FFN eigenspectra high-dimensional and near-isotropic, a property also shown in a recent line of work (Wang et al., 2026; Vasudeva et al., 2026), so the FFN nonlinearity does not need to substantially restructure representations. Dion follows the same trend but less strongly: PR gains are moderate and JS is higher and more uniform across layers, suggesting broader activation-driven reshaping than Muon yet still far less mismatch than AdamW. Thus, geometric updates serve as spectral equalizers, preventing the pre-activation spectrum from drifting into regimes that demand large nonlinear correction.

**The early-layer spectral collapse in AdamW forces FFN nonlinearity into repair mode.**

In contrast, AdamW exhibits very large PR(Post/Pre) gains concentrated in early layers, indicating optimizer-induced pre-activation collapse, as energy concentrating into a small set of dominant eigenmodes followed by strong nonlinear repair (Figure 9). However, this repair does not translate into better utilization, as the $\text{PR}_{\text{post}}$ remains lower than Muon and Dion in early and intermediate layers despite their massive PR gains during training. Thus, under AdamW the activation expends capacity primarily to undo collapse rather than refine a healthy spectrum, consistent with AdamW's worse perplexity (see Table 13).

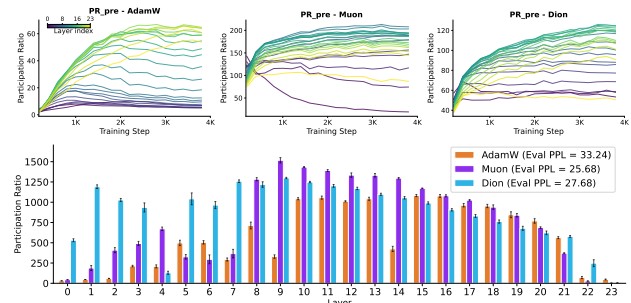

Figure 9: Layerwise PR_pre (top) over training, and final PR_post per-layer (bottom) for AdamW, Muon, and Dion optimizer. Muon maintains the highest PR_pre across almost all layers, Dion is intermediate, and AdamW shows early-layer collapse. Moreover, Muon concentrates the largest effective dimensionality in middle FFNs.

**Muon concentrates representational capacity where it matters: the middle FFNs.** Figure 9 isolates where effective dimensionality ($\text{PR}_{\text{post}}$) accumulates after training. Muon achieves the highest $\text{PR}_{\text{post}}$ in intermediate FFNs; whereas, Dion inflates $\text{PR}_{\text{post}}$ mainly in early FFNs without yielding the best perplexity; instead, the perplexity ordering follows mid FFNs $\text{PR}_{\text{post}}$ trend. This highlights the spectral mechanism for Muon's superiority—well-conditioned latent-space usage in the mid-FFNs.

The optimizer-dependent dynamics persist across scales and context lengths (Appendix L.1), and extend to Adafactor (Appendix L.2) and SGD on non-transformer MLP-Mixer (Appendix L.3). Thus, core findings—nonlinearity reinjects variance and flattens spectra—are optimizer-agnostic while remaining optimizer-dependent in degree, supporting the view that optimizers induce representational biases which should be used as explicit sources of inductive biases (Pascanu et al., 2025).

## 4 RELATED WORK

Prior work leverages spectral signals of weights or representations to understand model internals. RankMe (Garrido et al., 2023) and Diff-eRank (Wei et al., 2024) use spectral-entropy Rank measures to predict downstream accuracy and quantify compression. Bao et al. (2024) established the relation between spectral concentration of $QK$ weight matrix and attention localization, which Lee et al. (2025b) addressed using one-step belief-propagation refinement. Hu et al. (2025) showed that heavy-tail nature of weight ESD is biased by layer aspect ratio and propose fixed-aspect sub-ESD averaging to debias. Zhang et al. (2024) use JS divergence between Hessian spectra of parameter blocks in networks at initialization to suggest optimizer choice. Ruscio et al. (2025) used spectral gap and participation ratio on attention eigenspectra to reveal that attention sinks serve as geometric reference frames anchoring token representations. Poole et al. (2016) used Riemannian geometry with mean-field theory to demonstrate an order-to-chaos phase transition governed by the nonlinearity's derivative in randomly-initialized networks. Cowsik et al. (2025) used Lyapunov exponents for signal propagation, treating token evolution as a particle system to predict trainability. Dong et al. (2021) used relative residual norms and path decomposition to demonstrate attention-induced rank collapses.

In contrast, NerVE directly tracks FFN eigenspectrum dynamics, showing how nonlinearities redistribute variance in latent space, offering a diagnostic for architectural and optimizer choices.

## 5 LIMITATIONS AND CONCLUSION

While NERVE eigen-metrics analyze how FFNs organize variance in LLMs, they do not directly predict downstream task quality. Moreover, computing full eigendecompositions in large dimensions can be costly, often necessitating sampling or approximation (See Appendix G). Despite these constraints, our analysis shows that each metric contributes a distinct and complementary view of high-dimensional usage. Refer to Appendix M for further details on limitations of NerVE framework.

## REPRODUCIBILITY STATEMENT

We provide the training details, including hyperparameter configurations in the very-beginning of Section 3. The other implementation details are provided in Appendix A, including the project page.

## LLM USAGE STATEMENT

We utilized Large language models (LLMs) as a rewriting tool, mainly for grammar correctness and stylistic refinement, to improve the quality of written text. We also used LLMs for refining the plots, figures, and tables. We did not use LLMs for any intellectual contribution and experimental design.

## ACKNOWLEDGMENTS

This research was supported in part by the NSF CAREER Award #2340137 and a gift award from Google. We thank the anonymous reviewers for their valuable comments and constructive feedback, which significantly improved the quality of this work.

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

# A  IMPLEMENTATION DETAILS AND METHODOLOGICAL CONSIDERATIONS

## A.1  FRAMEWORK OVERVIEW

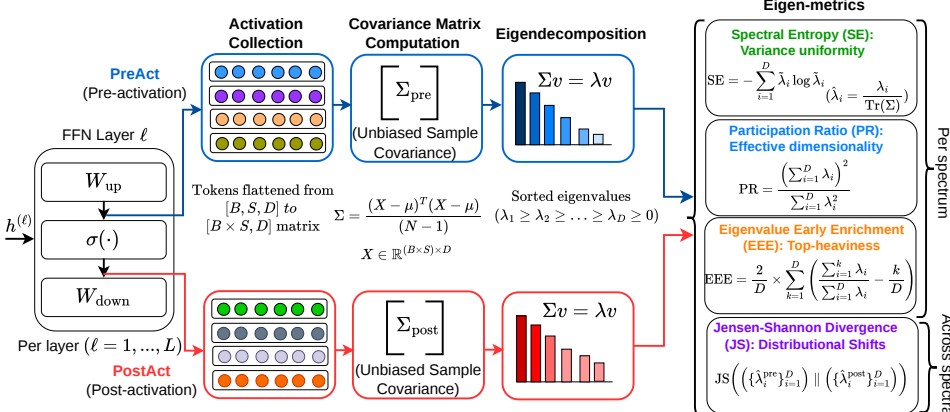

Figure 10: For each FFN layer $\ell$, we first collect *pre-activation* (after $W_{\text{up}}$, before $\sigma$) and *post-activation* (after $\sigma$, before $W_{\text{down}}$). Then, tokens are flattened into $X \in \mathbb{R}^{(B \times S) \times D}$ sample matrix, where $B$ is the global batch size, $S$ the sequence (context) length, and $D$ the FFN hidden dimension. Finally, we compute the unbiased sample co-variance of mean-centered activations to get the eigenvalues $\{\lambda_i\}_{i=1}^{D}$, and sorted them into descending order. Three eigen-metrics are computed on each (pre-/post-) eigenspectrum: spectral entropy (SE) for dispersion, participation ratio (PR) for effective dimensionality, and eigenvalue early enrichment (EEE) for top-heaviness; while Jensen-Shannon Divergence (JS) quantifies the distributional shift between the pre- and post-activation spectra, capturing the geometric restructuring performed by the nonlinearity.

## A.2  IMPLEMENTATION DETAILS FOR COMPUTING COVARIANCE MATRICES

During training, activations are collected through registered PyTorch hooks. For pre-activation, we use forward hooks on the output of the up-projection layer. For post-activation, we use pre-forward hooks on the down-projection layer to capture inputs before they enter the down-projection.

**Token aggregation.** Within each logging step, we flatten all tokens across the batch and sequence dimensions into a single matrix $X \in \mathbb{R}^{N \times D}$ where $N = B \times S$, treating each token embedding as an independent sample. We compute the mean-centered version $\hat{X} = X - \mu$ before forming $\Sigma = \frac{1}{N-1} \hat{X}^\top \hat{X}$. All eigenvalues are sorted in descending order ($\lambda_1 \geq \cdots \geq \lambda_D > 0$) before any metric computation—required for computing EEE since it depends on the cumulative sum from largest to smallest (Figure 10). **All results in this paper use the full-batch covariance** (no token sub-sampling). The effect of sub-sampling and low-rank approximation is discussed in Appendix G.

**Paired measurement.** Pre- and post-activation covariance matrices are computed on the identical set of $N$ tokens within each layer, ensuring that nonlinear transformations, including JS divergence which measures the geometric restructuring applied by the nonlinearity, are computed on the same input population rather than comparing statistics from different samples. With full-batch computation this pairing is implicit; however, when sub-sampling is employed (Appendix G), the same token subset must be used for both measurement points.

Our implementation ensures numerical stability and efficiency through:

1. *Precision control*: Converting all tensors to float32 to avoid precision issues.
2. *Numerical stability*: Adding a small epsilon values ($\epsilon = 1e - 12$) to prevent division by zero in eigenvalue normalization and entropy computation.
3. *Memory efficiency*: Processing layers sequentially and discarding intermediate covariance matrices after eigendecomposition, so that peak GPU memory is bounded by a single layer's covariance pair ($2D^2$ floats) rather than accumulating across all $L$ layers (see Appendix H.2). For distributed training, activations are gathered from all GPUs to rank 0 before covariance computation.

4. *Specialized computation*: Using `torch.linalg.eigvalsh` for symmetric positive semi-definite covariance matrices, which is both faster and more numerically stable.

This methodological rigor ensures that observed patterns in eigenvalue metrics reflect *intrinsic* properties of the network architecture rather than measurement artifacts. We provide the full implementation of our framework on project page: `https://nerve-eigenspectrum.github.io`

## B  EIGENSPECTRUM METRICS: DESIGN PRINCIPLES AND DIAGNOSTIC GUIDE

Table 4 summarizes the four eigen-metrics, their inputs, bounds, and what each captures. The remainder of this appendix provides justifications for NERVE eigen-metrics set (B.1), examines inter-metric relationships and failure modes with a diagnostic reference (B.2), and decomposes the geometric work of the nonlinearity into interpretable axes (B.3).

Table 4: Summary of the four NERVE eigen-metrics. Input indicates whether the metric operates on raw ($\lambda$) or normalized ($\hat{\lambda}$) eigenvalues, and range shows their (bounded) output values. SE, PR, and EEE characterize a single spectrum; while JS quantifies the divergence between the pre- and post-activation spectral shape. All four metrics are invariant to uniform scaling of eigenvalues.

| Metric | Input | Range | Sensitive to | Captures |
|---|---|---|---|---|
| SE | $\hat{\lambda}$ (normalized) | $[0, \ln D]$ | Full spectrum (especially mid-to-tail) | Variance uniformity (dispersion) |
| PR | $\lambda$ (raw) | $[1, D]$ | Large eigenvalues (suppresses small eigenvalues) | Effective dimensionality |
| EEE | $\lambda$ (raw) | $[0, 1)$ | Top eigenvalues (front-loaded concentration) | Top-heaviness (anisotropy) |
| JS | $\hat{\lambda}_{\text{pre}}$, $\hat{\lambda}_{\text{post}}$ | $[0, \ln 2]$ | Shape difference between (pre-/post-)spectra | Geometric restructuring |

### B.1  METRICS DESIGN AND JUSTIFICATIONS

The metrics should jointly capture distinct geometric aspects of high-dimensional latent space–no single metric suffices. Thus, our selection of eigen-metrics (SE, PR, EEE, JS) are based on the following desiderata for a diagnostic framework applied to high-dimensional eigenspectra:

(i) **Coverage.** Different eigenvalue distributions can share the same SE and PR, yet allocate variance very differently across the eigenspectrum. For instance, a spectrum with moderate variance spread uniformly across eigenmods, and a spectrum with the same total spread but concentrated in a few dominant directions followed by a broad low-variance tail could yield identical SE and PR values. *EEE resolves this ambiguity* by accounting cumulative variance of spectra, and able to differentiate the spectra having different fractions of latent-space utilization (Marbut et al., 2023).

(ii) **Complementary sensitivity.** The metrics in Table 4 emphasize different regions of the eigen-spectrum. SE, due to the logarithmic weighting to the normalized eigenvalues ($\hat{\lambda}_i$), is relatively *less sensitive to dominant modes* and more responsive to mid-to-tail spectrum. In contrast, PR downweights small eigenvalues and is driven primarily by dominant modes through the quadratic term ($\sum_i \lambda_i^2$). As a result, shifts confined to the tail can change SE while leaving PR nearly unchanged, whereas redistribution among leading eigenvalues can move PR strongly with only minor changes in SE. Since EEE summarizes top-heaviness of cumulative variance, a handful of large eigenvalues drives EEE close to 1, even if the rest are moderate. These complementary sensitivities ensure that changes in any region of the spectrum are reflected by at least one metric.

(iii) **Boundedness.** All four metrics have closed-form bounds (Table 4), enabling meaningful comparisons across models with different dimensionality $D$, training checkpoints, and architectural configurations. Unbounded metrics could make cross-configuration comparison unreliable.

(iv) **Scale invariance.** All four metrics are invariant to uniform scaling of the eigenvalues $\{\lambda_i\}_{i=1}^D$. This is crucial in practice, as the magnitude of the covariance matrix may fluctuate due to various factors such as batch statistics, or learning rate schedules. Scale invariance ensures the metrics remain focused on the *shape* of the distribution, which truly governs the directionality in the latent space. Moreover, each metric accounts for the entire eigenvalue distribution, unlike simple measures such as the eigenvalue ratio (largest-to-smallest), which are sensitive only to extremes.

(v) **Pre-post restructuring.** While SE, PR, and EEE characterize the intrinsic properties of a single distribution, JS quantifies the *information-theoretic distance* between two distributions (Nielsen & Nock, 2015). Thus, JS is needed to quantify nonlinearity-induced geometric restructuring. Large JS indicate significant variance redistribution across principal components, potentially creating

new directions of specialization or eliminating others. Conversely, small JS highlights FFNs which rescale existing directions without fundamentally altering the latent space geometry.

**Why Jensen-Shannon divergence over KL divergence** The main benefits of JS divergence (Eq. 6), compared to a simpler KL divergence, for eigenspectrum analysis are:

1. *Symmetry*: $\text{JS}(P_{\text{pre}} \parallel P_{\text{post}}) = \text{JS}(P_{\text{post}} \parallel P_{\text{pre}})$ enables unbiased comparison of pre-activation and post-activation eigenspectrum (Briët & Harremoës, 2009), whereas KL is asymmetric and would prioritize one distribution as the reference.

2. *Boundedness and interpretability*: $0 \leq \text{JS}(P_{\text{pre}} \parallel P_{\text{post}}) \leq ln(2)$, facilitating standardized comparisons across layers of different dimensions, and bounded scale makes JS values interpretable under distributional shift, unlike KL which could yields unbounded values in isolation.

3. *Numerical stability*: JS offers superior numerical stability when analyzing eigenspectra with near-zero eigenvalues, which are common in neural network representations.

### B.2 METRIC RELATIONSHIPS AND FAILURE MODES

While the four metrics are selected to capture distinct spectral properties, *they are not statistically independent*. In many training regimes, SE and PR move together because both respond to overall spectral flattening. Understanding when the metrics agree, when they diverge, and when individual metrics can mislead is crucial for reliable diagnostics.

**When metrics agree.** During standard training of well-configured models (e.g., GPT-2 PreLN with GELU), SE and PR typically co-increase while EEE decreases, highlighting nonlinearity-induced rank inflation through variance reinjection into previously inactive directions. JS decreases across depth, especially in deeper layers, indicating lesser geometric restructuring in deeper FFNs. When all four metrics align in this pattern, the diagnostic is straightforward, *the FFN latent space is being utilized effectively*.

**When metrics diverge, and why it matters?** Divergence between metrics signals that the spectral change is *localized* to a particular part of the eigenspectrum, which is diagnostically informative:

(i) *SE rises but PR is stable:* Variance is redistributing in the mid-to-tail eigenvalues without affecting the top eigenvalues that dominate PR. This happens when nonlinearity reshapes the spectral tail without shifting the dominant directions.

(ii) *PR rises but EEE also rises:* This paradoxical pattern occurs when a few new directions gain substantial variance while the rest remain near zero. PR increases because there are more active directions, but EEE increases because the new directions are still much larger than the inactive ones. This signals the trianing stages where network is allocating capacity to new features without distributing variance uniformly.

(iii) *SE and PR rise but JS is near zero:* Both pre- and post-activation spectra are flattening, but the nonlinearity is not reshaping the spectrum. This acts as a linear scaler which suggests the FFN may not be fully leveraging its nonlinear capacity.

**When individual metrics can mislead.** In the following scenarios, relying on a single eigen-metric could lead to misinterpretation of underlying learning dynamics.

(i) *PR without EEE can mislead on performance:* Hyperspherical normalization when used in FFN weights of a norm-free model, it achieves the highest PR_post (exceeding 600) but worse perplexity than Spectral normalization (Section 3.3), as their $\Delta$EEE flattening does not persist, EEE_post remains high across depth.

(ii) *PR in bottleneck regimes:* At FFN width $D = 1d$ (no dimensional expansion), PR is constrained to a narrow range regardless of spectral quality. PR values in this regime should not be compared directly to wider configurations without normalization by $D$.

(iii) *SE blind to absolute scale:* Since SE operates on normalized eigenvalues ($\hat{\lambda}_i = \lambda_i / \text{Tr}(\Sigma)$), it is insensitive to the to absolute scale, and it should be used to locate the spectral bottlenecks (layers with very low SE, See Figure 11). Nonetheless, two FFNs with identical spectral shapes but substantially different total variance ($\text{Tr}(\Sigma)$) would result in identical SE values. Thus, when total variance matters, for instance comparing different layers within a network (activation magnitudes vary substantially across depth), **PR should be preferred over SE**.

(iv) *EEE saturation:* In highly anisotropic spectra (e.g., a single dominant eigenvalue with near-zero rest), EEE saturates near its upper bound and becomes insensitive to further concentration. In this regime, PR (which continues to decrease toward 1) is the more informative metric.

**Diagnostic references.** Table 5 summarizes the joint metric signatures most frequently observed across our experiments, along with their interpretation. To avoid misinterpretation, it is important to report all four metrics jointly.

Table 5: Diagnostic reference for joint eigen-metric signatures. Each row describes a commonly observed pattern, its interpretation, and where in the paper it is empirically demonstrated.

| Joint Signatures | Interpretation | References |
|---|---|---|
| SE↑ PR↑ EEE↓ | Healthy spectral flattening (Rank inflation) | Standard training (§3.1) |
| SE↓ PR↓ EEE↑ | Spectral collapse | AdamW pre-activation (§3.7) |
| JS↓ across depth | Decreasing nonlinear restructuring | Healthy deep layers (§3.1) |
| SE↑ PR↑↑ JS↓ across depth | Nonlinearity compensating for pathology | Norm-free ReLU models (§3.2) |
| EEE ↓↓ JS↑ across depth | Sustained spectral flattening | Spectral norm. in FFNs (§3.3) |
| Pre-act stable, post-act shifts | Repair regime | AdamW (§3.7 & Appendix L.1) |
| Pre-act shifts, post-act stable | Refinement regime | Muon (§3.7 & Appendix L.1) |

### B.3 DECOMPOSING NONLINEARITY-INDUCED EIGENSPECTRUM RESTRUCTURING

In addition to JS metric, we use two derived cross-activation quantities: the participation ratio gain PR(Post/Pre) and the EEE difference $\Delta EEE = EEE\_post - EEE\_pre$. These quantities decompose what the nonlinearity does to the eigenspectrum:

- **JS** measures the *total magnitude* of spectral change, without indicating direction.

- **PR(Post/Pre)** measures *dimensional expansion*, whether nonlinearity activated new directions.

- **$\Delta$EEE** measures *top-heaviness reduction*, whether the nonlinearity suppressed leading eigenvalue dominance. More negative $\Delta$EEE indicates stronger flattening.

These metrics are complementary, not redundant: dominant-mode redistributions can raise JS divergence with little PR gain, whereas tail spreading across many small modes can yield large PR gains with a typically more moderate increase in JS divergence.

**Effort does not imply outcome.** Large PR gain does not imply high $PR_{post}$. Under AdamW (Section 3.7), nonlinearities yield the largest PR gains and highest JS, yet the lowest $PR_{post}$ (vs. Muon/Dion). In contrast, Muon attains the highest $PR_{post}$ with the smallest PR gains and lowest JS. The difference is that AdamW collapses the pre-activation spectrum, so the nonlinearity expends capacity on *repair* (large corrective gains) that only recover a mediocre state, whereas Muon preserves a well-conditioned pre-spectrum and requires only modest *refinement*.

**Sustained flattening matters.** High $PR_{post}$ alone is not sufficient: HNorm (Section 3.3) attains larger $PR_{post}$ yet underperforms, whereas SNorm maintains consistently lower $EEE_{post}$ throughout training (sustained flattening) and achieves best perplexity among weight-normalization methods.

**Characteristic patterns.** Table 6 summarizes four diagnostically distinct combinations observed across our experiments, each reflecting a qualitatively different regime of nonlinear function.

Table 6: Four regimes of nonlinear geometric work, distinguished by the joint signature of JS, PR(Post/Pre), and $\Delta$EEE. JS alone cannot distinguish beneficial restructuring from compensatory repair; PR gain alone cannot distinguish beneficial restructuring from expansion without equalization.

| JS | PR gain | $\Delta$EEE | Interpretation |
|---|---|---|---|
| High | High | Strongly negative | Beneficial Restructuring
Active expansion and flattening (norm-free ReLU, §3.2) |
| High | High | Weakly negative | Compensatory Repair
Effort wasted undoing PreAct spectral collapse (AdamW, §3.7) |
| Moderate | High | Near zero or positive | Expansion without equalization
New directions but top-heaviness persists (HNorm, §3.3) |
| $\approx 0$ | Moderate | $\approx 0$ | Spectral Inertia
Nonlinearity acts as near-identity (norm-free GELU, §3.2) |

## C  EIGENSPECTRAL SIGNATURE

### C.1  SPECTRAL SIGNATURE OF LAYERNORM POSITIONING: PRELN, MIXLN, AND POSTLN

Figure 11 illustrates the post-activation spectral signatures (SE_post and PR_post) for three LayerNorm placements—PreLN, PostLN, and MixLN—across GPT2-125M, LLaMA-70M, and LLaMA-130M. These signatures highlight the extent of FFN latent space utilization across layers. Within each model family, the ranking of evaluation perplexities is corroborated by their spectral signatures: models with lower perplexity exhibit higher utilization.

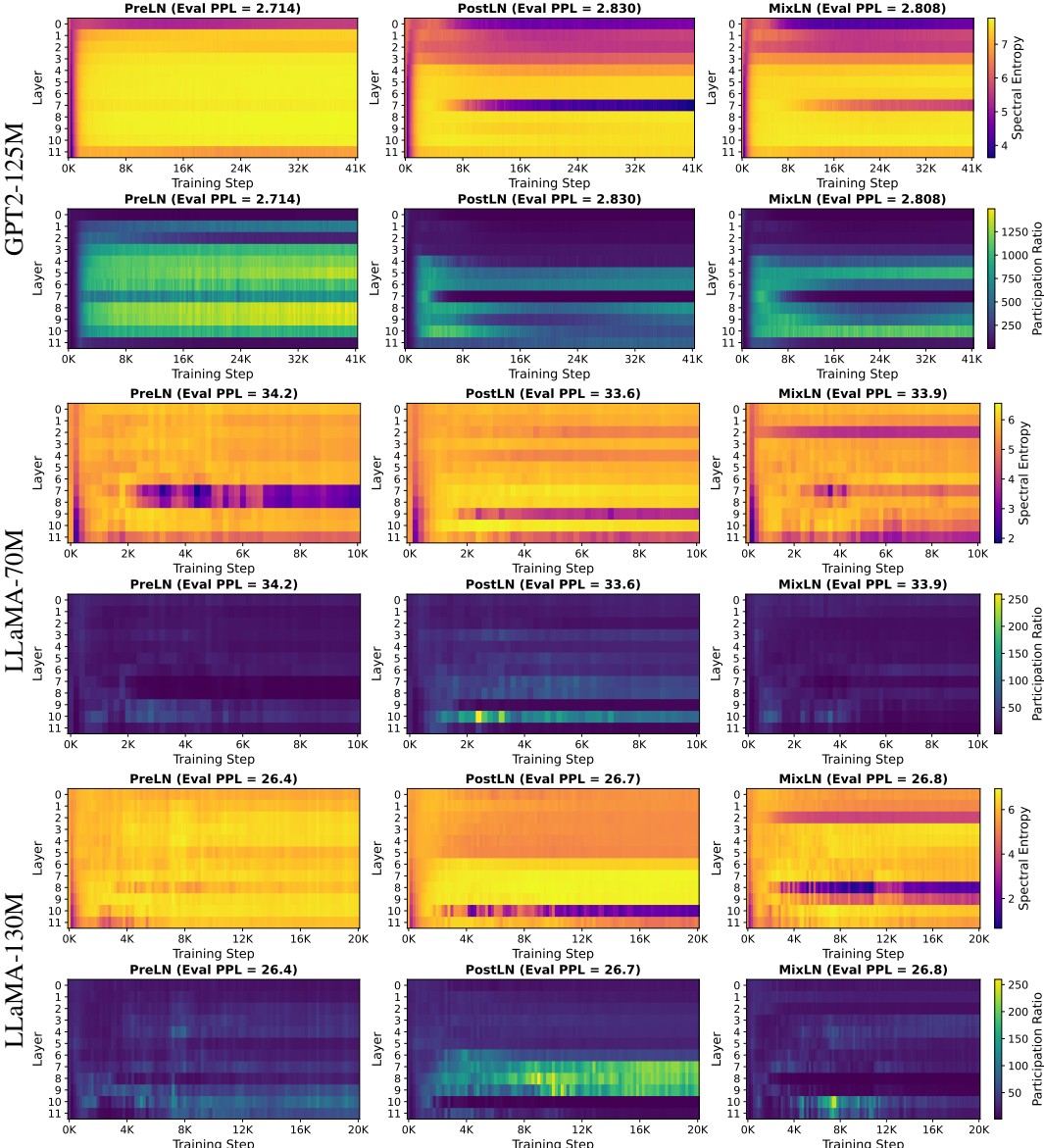

Figure 11: Eigenspectral impact of LayerNorm placement (PreLN, PostLN, MixLN) in GPT-2 and LLaMA variants (70M, 130M). The spectral signatures are shown through post-activation spectral entropy (↑) and participation ratio (↑), and model's perplexity is shown on top of each plots. GPT-2 models trained on CodeParrot and LLaMA variants on C4.

**GPT2-125M:** Performance follows the order PreLN (PPL = 2.714) > MixLN (2.808) > PostLN (2.830). The spectral signatures follow this ranking: PreLN exhibits superior spectral entropy and participation ratio trend across layers, indicating more effective FFN latent space utilization, while PostLN shows the most constrained spectral characteristics. In particular, $L7$ in PostLN and MixLN show very-low utilization compared to the PreLN configuration.

**LLaMA-70M:** PostLN achieves the lowest perplexity (PPL = 33.6), followed by MixLN (33.9), while PreLN performs worst (34.2). The eigenspectral analysis shows that PreLN exhibits substantially lower spectral entropy in deeper layers ($L7$-$L8$) compared to PostLN and MixLN. In contrast, PostLN demonstrates superior spectral characteristics with higher participation ratios in deeper layers compared to MixLN, consistent with its lower perplexity.

**LLaMA-130M:** PreLN yields the best performance with a perplexity of 26.4, followed closely by PostLN (26.7) and MixLN (26.8). While PostLN exhibits stronger spectral entropy and participation ratio in the mid-depth layers ($L6$-$L9$), PreLN consistently outperforms across the remaining layers, particularly $L10$ where PostLN deteriorates, undermining its overall benefits. In contrast, MixLN shows the worse spectral profile, leads to highest perplexity.

## C.2   Spectral Signature in Larger LLaMA Models

Figure 12 shows the post-activation eigen-spectral signatures (SE_post and PR_post) for LayerNorm placements, PreLN and MixLN, in LLaMA-250M and LLaMA-1.3B. Note that PostLN is excluded since it becomes unstable at these larger scales (Li et al., 2025). These spectral signatures provide a quantitative assessment for latent space utilization in each FFNs.

**LLaMA-250M:** MixLN outperforms the PreLN configuration by a margin of 0.3 in terms of PPL improvement (24.2 vs 24.5). The spectral signature of PreLN exhibits a consistent lower SE in layer 7 to 16, and lower PR in mid-depth regions (Figure 12). Whereas, MixLN avoid this mid-network spectral collapse pattern by confining the lower SE bands in early layers while maintaining higher SE and PR through the mid-depth layers, *effectively placing most of the usable capacity where the model does the bulk of its computation*. This redistribution of spectral entropy and participation ratio across depth is consistent with the lower perplexity achieved by the MixLN model.

**LLaMA-1.3B:** At 1.3B scale, MixLN's strategic LayerNorm placement unable to facilitate favorable capacity distribution, and training collapses after 7K steps. Whereas, PreLN maintain a stable spectral entropy across layers and achieves notably higher participation ratio in the deeper layers throughout training. This divergence in spectral behavior aligns with the final performance gap, as MixLN's perplexity explodes to 1457.1 compared to 21.2 for PreLN.

These results suggest that LayerNorm placement significantly influences how effectively the FFN utilizes its latent space. Pre-LN configurations appear to better preserve and amplify feature diversity, especially in deeper layers, thereby enabling higher-dimensional latent representations. The observed gains in the MixLN setup indicate that even partial use of PreLN can compensate for the limitations of PostLN, offering a potential strategy for balancing stability and expressivity.

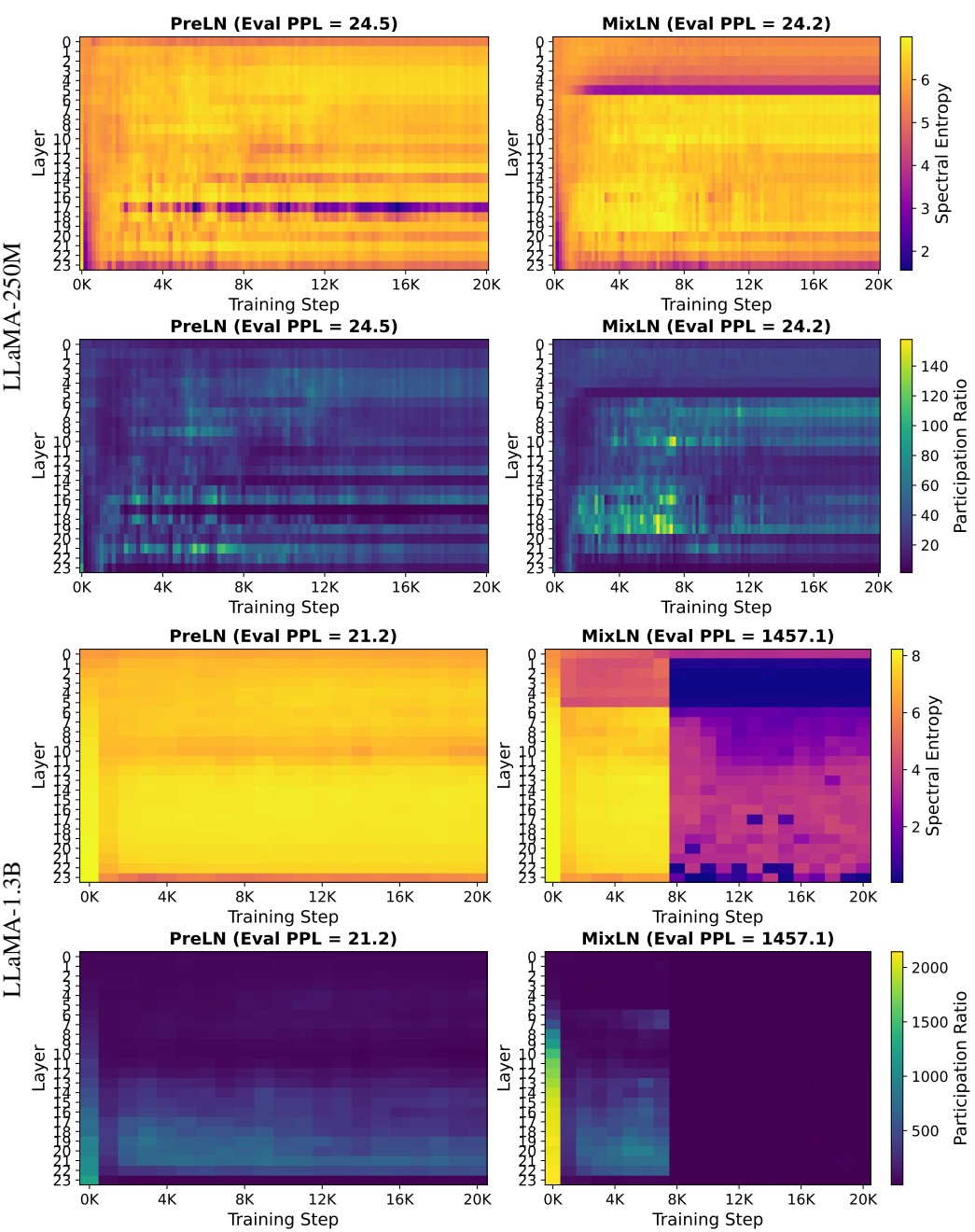

Figure 12: Eigenspectral impact of LayerNorm placement (PreLN vs MixLN) in LLaMA-250 and LLaMA-1.3B models, trained from scratch on C4 dataset for 20K iterations with 256 context length. The spectral signatures are shown through SE_post (↑) and PR_post (↑), and model's perplexity is shown on top of each plots. MixLN variant of LLaMA-1.3B destabilized after 7K iterations.

## C.3 Spectral Signature of Positional Encoding: NoPE vs RoPE

Figure 13 shows the spectral signature of rotary positional encoding (RoPE), in contrast with no positional encoding (NoPE). In particular, the layerwise spectral entropy and EEE values of post-activation eigenspectrum are shown.

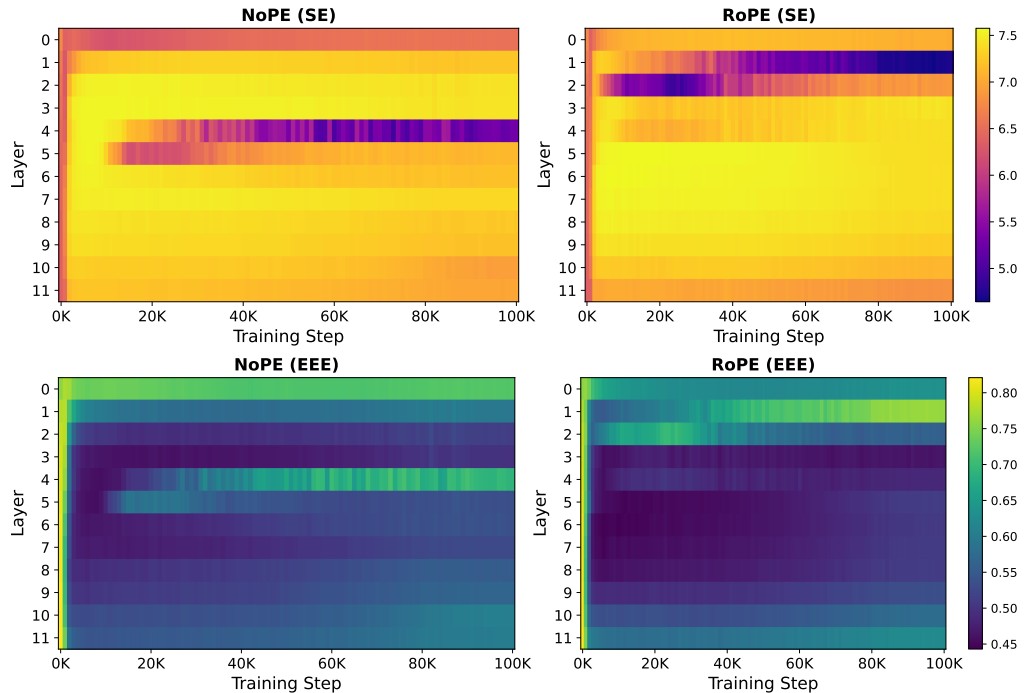

Figure 13: Spectral Signature of Positional Encoding (RoPE vs NoPE) in GPT-2 models trained with 512 context size on 26B token form openwebtext dataset

## D LayerNorm Positioning and FFN Width Sweep

Section 3.4 reports the normalized PR_post (PR/$D$) to compare the *efficiency* of width utilization across LayerNorm configurations. Table 7 complements this with absolute PR$_{post}$ values, revealing the scale of the gap that the normalized view can mask. At $D = 6144$, PreLN sustains PR$_{post} \approx 1822$ effective dimensions, compared to 71 for PostLN, a $25\times$ difference in absolute latent capacity that Figure 6 can understate. MixLN remains closer to PostLN throughout the width sweep, fluctuating between PR$_{post} \approx 137$–278 without scaling proportionally with $D$, suggesting that neither MixLN nor PostLN converts additional width into usable dimensions as effectively as PreLN.

Table 7: PR_post (median $\pm$ MAD across 12 layers) at final training checkpoint for different LayerNorm placements and FFN widths in GPT-2 (125M).

| Method | D=768 | D=1536 | D=2048 | D=3072 | D=3840 | D=4608 | D=5376 | D=6144 |
|---|---|---|---|---|---|---|---|---|
| PreLN | 292±62 | 573±149 | 831±238 | 1002±354 | 1275±333 | 1562±429 | 1721±244 | 1822±370 |
| MixLN | 137±111 | 184±126 | 247±233 | 274±224 | 278±234 | 159±142 | 251±201 | 233±187 |
| PostLN | 95±65 | 142±104 | 151±147 | 240±231 | 117±110 | 91±83 | 94±84 | 71±62 |

# E    SPECTRAL SIGNATURE ACROSS FFN WIDTH SWEEPS BASELINE MODELS

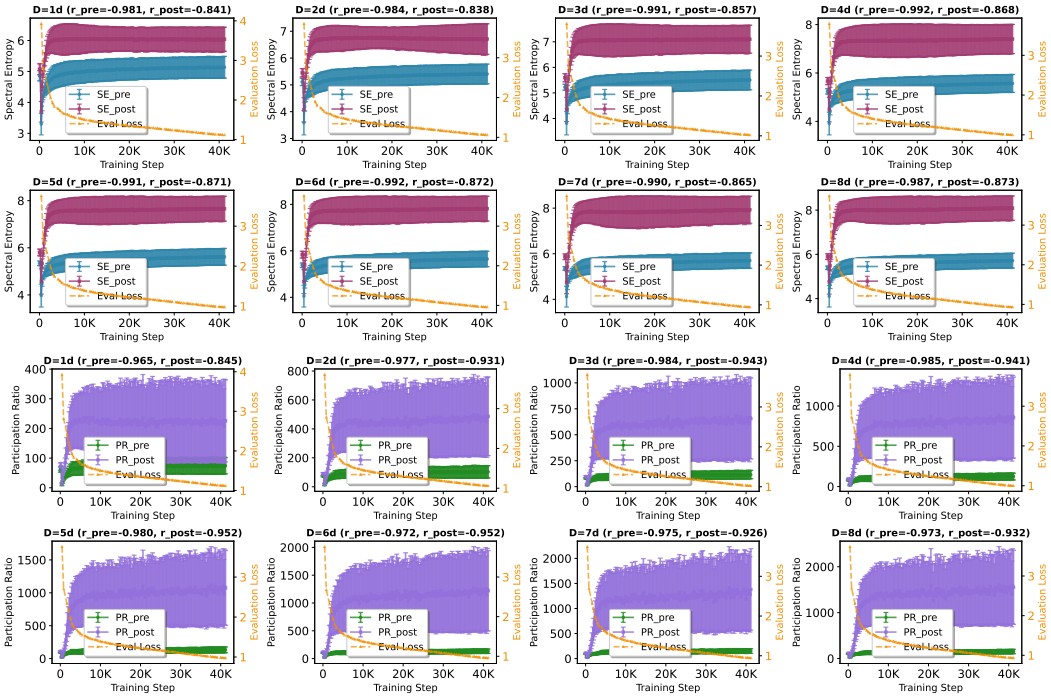

Figure 14: Eigen metrics in GPT-2 (**GELU**, D=$1d$ to $8d$) with Pearson $r$ to eval loss ($r_{\mathrm{pre}}, r_{\mathrm{post}}$)

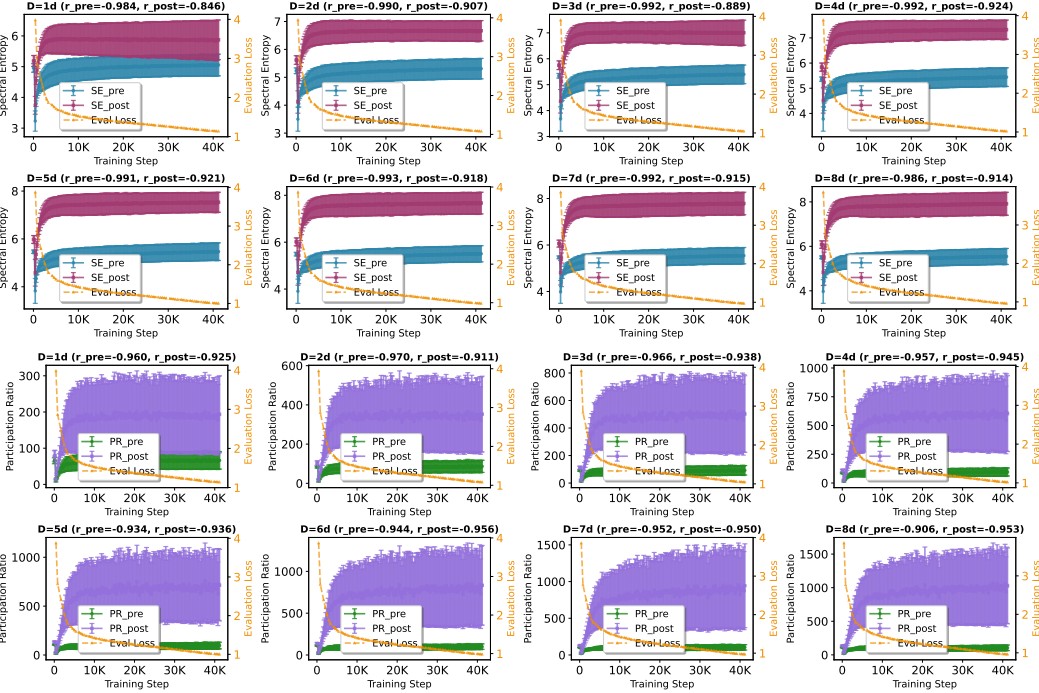

Figure 15: Eigen metrics in GPT-2 (**ReLU**, D=$1d$ to $8d$) with Pearson $r$ to eval loss ($r_{\mathrm{pre}}, r_{\mathrm{post}}$)

# F SPECTRAL SIGNATURE ACROSS FFN WIDTH SWEEPS IN NORM-FREE

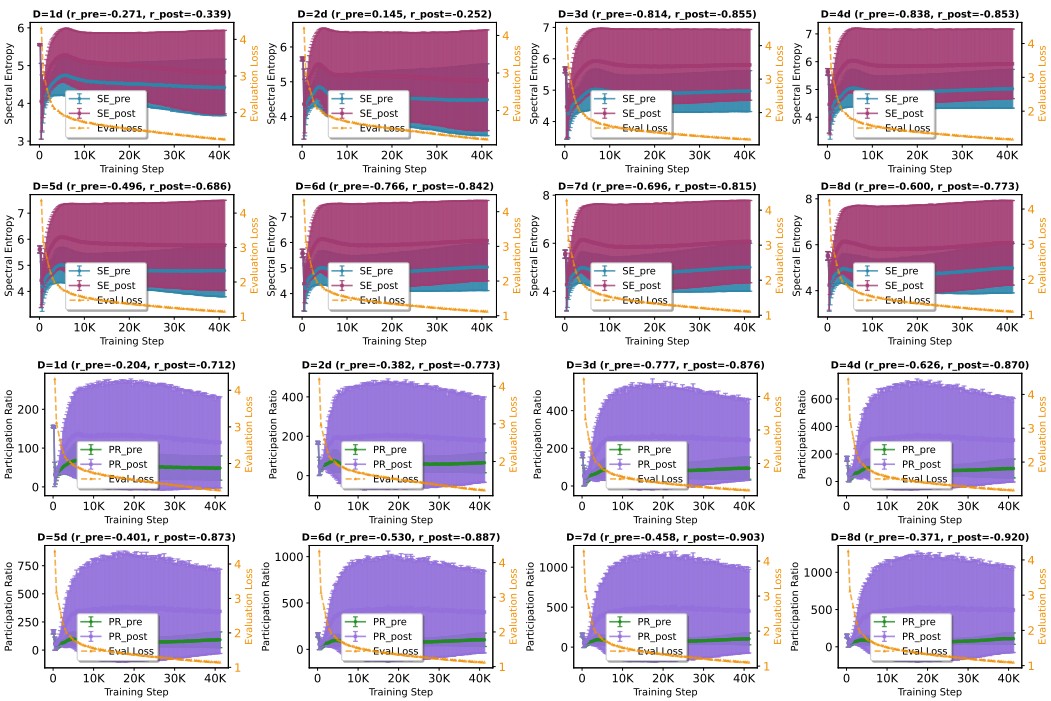

Figure 16: SE and PR in **Normalization-free** GPT-2 (**GELU**, D=1$d$ to 8$d$) with Pearson correlation to eval loss ($r_{\mathrm{pre}}, r_{\mathrm{post}}$)

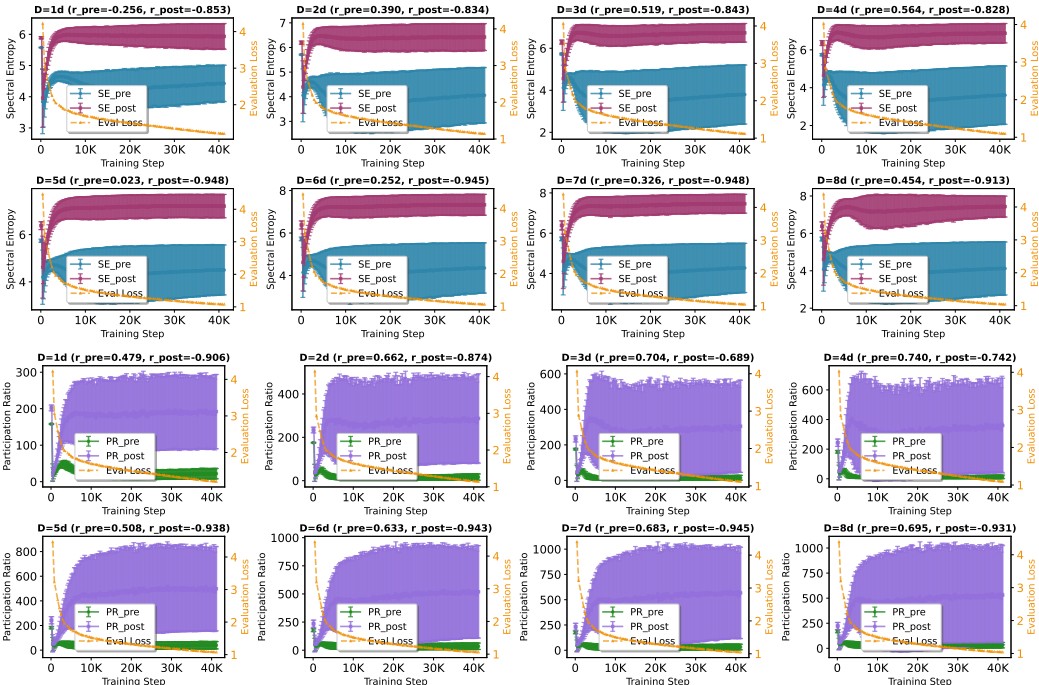

Figure 17: SE and PR in **Normalization-free** GPT-2 (**ReLU**, D=1$d$ to 8$d$) with Pearson correlation to eval loss ($r_{\mathrm{pre}}, r_{\mathrm{post}}$)

## G    TOKEN SUB-SAMPLING AND LOW-RANK APPROXIMATION

Full-batch covariance computation is exact but becomes costly at scale (see Appendix H). We evaluate two approximation strategies: token-level sub-sampling (using 5%, 10%, 25%, and 50% of tokens) and low-rank eigendecomposition. For the latter, we use RandSVD (Halko et al., 2011), which uses randomized projections (optionally with power iterations) to approximate the top-$k$ eigenspectrum, and Lanczos iteration (Lanczos, 1950). We evaluate both at ranks $k \in \{256, 512\}$, covering two widely used paradigms for scalable eigendecomposition (Ghorbani et al., 2019).

### G.1    EIGEN-METRIC FIDELITY AND DIAGNOSTIC VALIDITY UNDER APPROXIMATIONS

**Token sub-sampling better preserves metric fidelity than low-rank truncation** Table 8 reports the percentage error relative to full-batch eigendecomposition for GPT-2 (GELU). Token sub-sampling preserves all four metrics with minimal distortion—at 10% sampling, the worst-case error is 10.24% (PR_post), with SE and EEE errors below 2%. By contrast, low-rank methods introduce substantial bias: because SE, PR, EEE, and JS aggregate information across the full spectrum, truncating tail eigenvalues systematically distorts the metrics, with PR_post errors exceeding 90% even at rank 512. Therefore, token sub-sampling should be used over low-rank approximation when computational constraints require approximate eigendecomposition.

Table 8: Percentage of error when sampling and low-rank approximation methods are used in GPT-2.

| Metric | Sampling | | | | RandSVD | | Lanczos | |
|---|---|---|---|---|---|---|---|---|
| | 5% | 10% | 25% | 50% | 256 | 512 | 256 | 512 |
| SE_pre (%) | 1.0 | 0.5 | 0.5 | 0.4 | 50.7 | 50.1 | 7.9 | 1.7 |
| SE_post (%) | 4.8 | 1.9 | 1.3 | 1.3 | 63.7 | 61.0 | 28.8 | 19.3 |
| PR_pre (%) | 4.9 | 4.1 | 5.1 | 4.3 | 90.4 | 90.2 | 18.0 | 3.6 |
| PR_post (%) | 16.7 | 10.2 | 33.5 | 30.2 | 94.9 | 94.8 | 68.4 | 53.9 |
| EEE_pre (%) | 0.4 | 0.1 | 0.1 | 0.1 | 4.2 | 1.6 | 38.8 | 26.3 |
| EEE_post (%) | 20.7 | 8.6 | 1.4 | 1.2 | 35.7 | 38.6 | 30.5 | 28.8 |
| JS (%) | 21.5 | 8.3 | 3.0 | 3.4 | 67.5 | 63.2 | 81.1 | 70.5 |

**Approximation preserves pre-activation but not post-activation diagnostic power.** Low absolute error in a metric does not necessarily preserve its *diagnostic utility*: approximations can alter the rank-ordering of configurations and weaken their correlation with network's generalization performance. Table 9 evaluates this correlations between each eigen-metric and validation loss under approximation.

Table 9: Effect of token-sampling and low-rank approximation on metric–(eval)loss correlation, evaluated on GPT-2 (125M). Token-sampling preserves pre-metric trends but degrades post-metric correlations since tail eigenvalues are under-sampled; low-rank truncation distorts both.

| Metric | Sampling | | | | RandSVD | | Lanczos | |
|---|---|---|---|---|---|---|---|---|
| | 5% | 10% | 25% | 50% | 256 | 512 | 256 | 512 |
| SE_pre | -0.97 | -0.972 | -0.971 | -0.972 | -0.106 | -0.174 | -0.822 | -0.959 |
| SE_post | -0.34 | -0.376 | -0.332 | -0.363 | 0.612 | 0.568 | 0.335 | 0.08 |
| PR_pre | -0.914 | -0.915 | -0.918 | -0.912 | 0.014 | 0.047 | -0.788 | -0.901 |
| PR_post | -0.122 | -0.138 | 0.049 | -0.235 | 0.594 | 0.575 | 0.314 | 0.136 |

Token sub-sampling largely preserves *pre-activation* correlations: $\text{SE}_{\text{pre}}$ remains $|r| > 0.97$ and $\text{PR}_{\text{pre}}$ remains $|r| > 0.91$ across sampling ratios (5%-50%). In contrast, *post-activation* correlations are less stable under sub-sampling ($\text{SE}_{\text{post}}$ drops to $|r| \approx 0.34$–$0.38$ and $\text{PR}_{\text{post}}$ to $|r| < 0.24$, with a sign flip at 25%), suggesting that sub-sampling noise disproportionately affects the **mid-to-tail spectrum**.

Low-rank approximations degrade correlations more severely: RandSVD yields weak correlations for both pre- and post-activation metrics ($\text{SE}_{\text{pre}}\ |r| < 0.18$, $\text{PR}_{\text{pre}}\ |r| < 0.05$). Lanczos performs better for pre-activation metrics (e.g., Lanczos-512: $\text{SE}_{\text{pre}}\ |r| = 0.96$, $\text{PR}_{\text{pre}}\ |r| = 0.90$), but post-activation correlations remain low ($\text{SE}_{\text{post}}\ |r| < 0.34$, $\text{PR}_{\text{post}}\ |r| < 0.14$). Overall, token sub-sampling at $\geq 10\%$ preserves most pre-activation diagnostic power with minimal loss, whereas correlation-based analyses of post-activation metrics benefit from full-batch covariance estimation ($|r| > 0.84$, Table 3). Low-rank truncation is unsuitable when preserving metric–loss correlations is the primary objective.

## G.2 SPECTRAL SIGNATURE OF SAMPLING AND LOW RANK APPROXIMATION

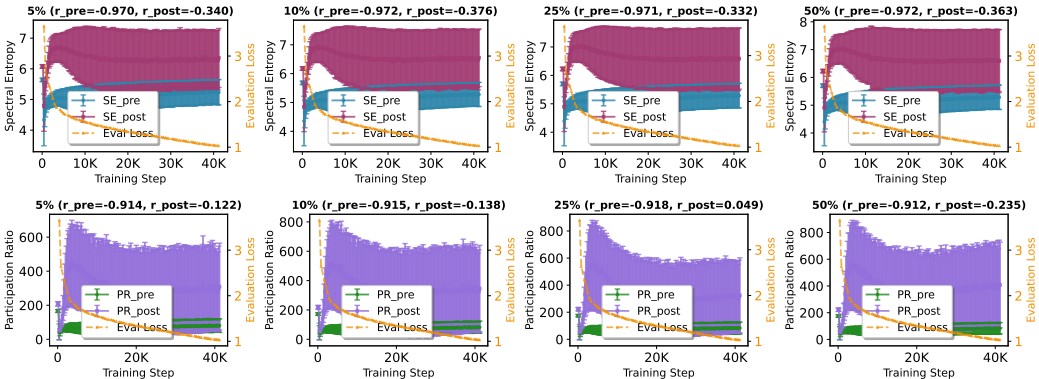

Figure 18: SE and PR in GPT-2 with Pearson $r$ to eval loss under **sub-sampling** (5, 10, 25, 50%)

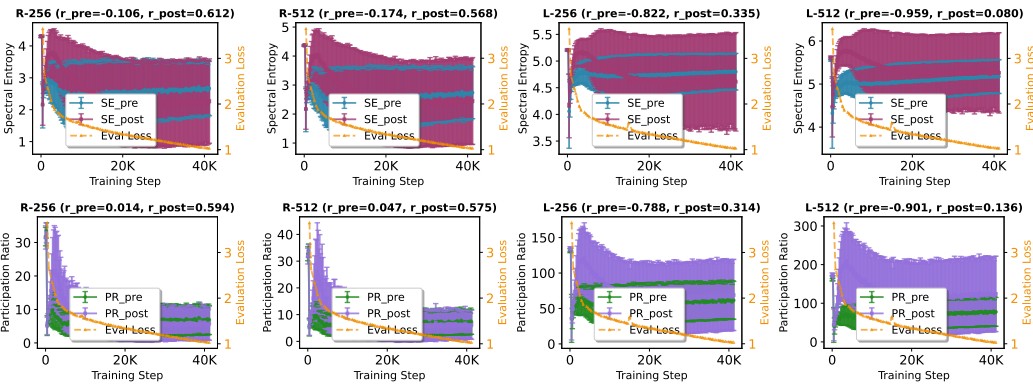

Figure 19: SE and PR in GPT-2 with Pearson $r$ to eval loss under **low-rank approximation**.

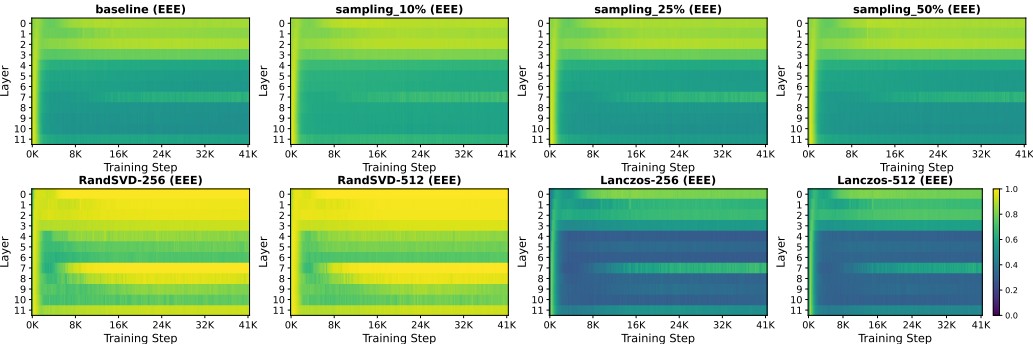

Figure 20: Impact of sampling (10%, 25%, 50%), and low-rank approximation on EEE (GPT-2)

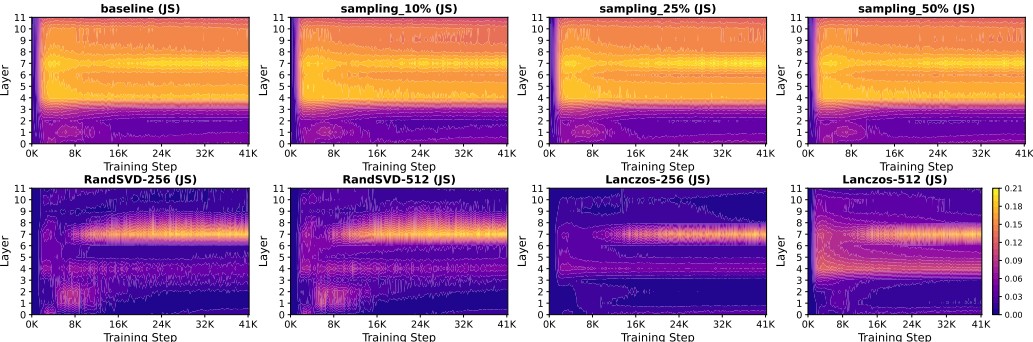

Figure 21: Impact of sampling (10%, 25%, 50%), and low-rank approximation on JS metric (GPT-2)

# H  COMPUTATIONAL AND MEMORY EFFICIENCY OF NERVE

## H.1  COMPUTATIONAL COMPLEXITY AND MEMORY OVERHEADS OF NERVE

Table 10 reports wall-clock time and relative overhead for computing eigenspectrum metrics on GPT-2 (a $3072 \times 3072$ covariance) across approximation methods. Full-batch eigendecomposition takes 14.41s per logging step, corresponding to a 6.4% overhead when logging every 200 steps, and 1.3% when logging every 1000 steps. Token sub-sampling at 5% reduces this to 9.89s, with wall-clock time increasing gradually from 9.89s to 14.41s (full batch). At the recommended logging frequency of every 1000 steps, all methods, including full batch, add approximately 1% overhead, making approximation irrelevant at this scale. Sub-sampling or low-rank methods become relevant only at substantially larger FFN dimensions ($D \gg 3072$).

Table 10: Wall-clock time and relative overhead for computing eigenspectrum metrics at various logging frequencies. Overhead is reported as percentage of total training time when eigendecomposition is performed every 200 and 1000 steps on GPT-2 with $3072 \times 3072$ FFN covariance matrix size running on AMD EPYC 7502 server with NVIDIA RTX 3090 GPU

| Metric | Sampling | | | | RandSVD | | Lanczos | | Full |
|---|---|---|---|---|---|---|---|---|---|
| | 5% | 10% | 25% | 50% | 256 | 512 | 256 | 512 | batch |
| Wall-clock time | 9.89s | 9.92s | 11.48s | 12.62s | 10.28s | 11.19s | 12.07s | 12.20s | 14.41s |
| Overhead-200 (%) | 4.37 | 4.39 | 5.08 | 5.58 | 4.55 | 4.95 | 5.34 | 5.40 | 6.38 |
| Overhead-1K (%) | 0.87 | 0.88 | 1.02 | 1.12 | 0.91 | 0.99 | 1.07 | 1.08 | 1.28 |

## H.2  MEMORY-EFFICIENT EIGENSPECTRUM ANALYSIS

To enable scalable eigenvalue analysis during training, we implemented memory optimization strategies, listed in Algorithm 1. As shown in Table 11, these memory optimization significantly reduces the peak GPU memory usage. For instance, for GPT-2 model (D=$4d$) the peak GPU memory usage is restricted to $\approx 2 \times 36$MB per layer rather than accumulating $2 \times 12 \times 36$MB $= 864$MB across all FFNs per logging step.

Table 11: GPU memory overhead for full-batch eigen-computation across various FFN widths.

| Metric | D=$1d$ | D=$2d$ | D=$3d$ | D=$4d$ | D=$5d$ | D=$6d$ | D=$7d$ | D=$8d$ |
|---|---|---|---|---|---|---|---|---|
| Matrix dim. | [768,768] | [1536, 1536] | [2304, 2304] | [3072, 3072] | [3840, 3840] | [4608, 4608] | [5376, 5376] | [6144, 6144] |
| Matrix size | 2.25MB | 9MB | 20.25MB | 36MB | 56.25MB | 81MB | 110.25MB | 144MB |
| GPU Mem. | 4.5MB | 18MB | 40.5MB | 72MB | 112.5MB | 162MB | 220.5MB | 288MB |

The complete pipeline's wall-clock time, following the steps in Algorithm 1, is measured as follows:

```
1  # Start timing (includes all operations below)
2  torch.cuda.synchronize()
3  start_time = time.time()
4
5  # Complete eigenvalue pipeline for all layers:
6  # 1. CPU-GPU data transfer of activations
7  # 2. Covariance matrix computation (3072x3072)
8  # 3. Eigenvalue decomposition using torch.linalg.eigvals()
9  # 4. Eigenvalue metrics computation (SE, PR, EEE, JS)
10 # 5. Memory cleanup and GPU cache clearing
11
12 torch.cuda.synchronize()
13 end_time = time.time()
14 overhead = end_time - start_time  # Complete wall-clock time
```

Listing 1: Wall-clock timing measurement for eigenspectrum-based computational overhead

To enable scalable eigenvalue analysis during training, we implement three memory optimization strategies that significantly reduce GPU memory overhead:

1. **Eigenvalue-only computation:** Since our eigen metrics depend only on eigenvalues, we employ `torch.linalg.eigvalsh` to compute eigenvalues without eigenvectors.

```
1 ### What we're using (efficient):
2 # Only eigenvalues: 3072 values
3 vals = torch.linalg.eigvalsh(cov)
4
5 ### Less efficient alternatives:
6 # Eigenvalues + eigenvectors: 3072 + (3072x3072)
7 vals, vecs = torch.linalg.eigh(cov)
8 # General eigendecomposition (even more overhead)
9 vals, vecs = torch.linalg.eig(cov)
```

Listing 2: Memory-efficient eigenvalue computation

2. **Sequential layer processing:** Rather than computing eigenvalues for all layers simultaneously, we process layers sequentially with memory cleanup between computations:

```
1 # Process each layer individually with cleanup
2 for layer_idx in sorted(self.layer_pre_acts.keys()):
3     # Compute metrics for current layer
4     # ...
5     self.layer_pre_acts[layer_idx].clear()
6     gc.collect()
7     torch.cuda.empty_cache()
```

Listing 3: Sequential layer processing with memory cleanup

This approach maintains peak GPU memory usage at $\sim 2\times36$MB per layer rather than accumulating $2\times12\times36$MB = 864MB across all FFNs per logging step (Table 11).

3. **Hybrid storage strategy:** We store activation tensors on CPU memory while performing eigenvalue computations on GPU, balancing memory efficiency with computational speed.

---

**Algorithm 1** Memory-Efficient Eigenspectrum Analysis

---

1: **for** layer $\ell = 1, \ldots, L$ **do**
2:      $\mathbf{H}_\ell \leftarrow \text{GetActivations}(\ell)$          ▷ Store on CPU
3:      $\boldsymbol{\lambda}_\ell \leftarrow \text{eigvalsh}(\mathbf{H}_\ell^\top \mathbf{H}_\ell / N)$          ▷ O(d) memory
4:      Compute metrics: SE, PR, EEE, JS from $\boldsymbol{\lambda}_\ell$
5:      **del** $\mathbf{H}_\ell$          ▷ Immediate cleanup
6: **end for**

---

# I   EIGENSPECTRUM DYNAMICS IN A NON-TRANSFORMER ARCHITECTURE

We selected MLP-Mixer (Tolstikhin et al., 2021) as a non-transformer architecture family model for three key reasons: 1) it has the core architectural block that we studies in this work, wider MLPs/FFNs with LayerNorm and GELU activations, while completely removing the self-attention block; 2) the channel-mixing MLPs/FFNs in MLP-Mixer are functionally analogous to FFNs in Transformers, as they expand and squeeze the latent representation with an intervening nonlinearity, and stacked across the network's depth; and 3) it isolate the contribution of FFN nonlinear transformations from attention-specific dynamics like rank collapse in self-attention (Dong et al., 2021).

Moreover, as the MLP-Mixer is designed for vision tasks, it has fundamentally different inductive biases compared to decode-only LLMs. More importantly, it allows us to extend the eigenspectrum analysis from language modeling to vision tasks, verifying the generalization across modalities. This lets us ask: *do the eigenspectrum patterns we observe depend fundamentally on the attention mechanism, or are they a more general property of deep feedforward layers?*

**Settings: Model, dataset, and training hyperparameters** We use MLP-Mixer B/16 models adapted for CIFAR-100 ($32\times32$ images, $4\times4$ patches, 64 tokens in one sequence) since it closely resembles the key architectural parameters from GPT-2 to enable direct comparison. The architecture consists of 12 Mixer blocks with embedding dimension 768. Each block contains two MLPs: token-mixing (hidden dimension 384) and channel-mixing (hidden dimension 3072), both using LayerNorm and GELU activation. This configuration results in 57.4M trainable parameters.

We train mixer models for 120 epochs using the Adam optimizer (Kingma, 2015) with a learning rate of $1e$-3 and weight decay of $5e$-5. We use a batch size of 128 and employ cosine annealing for learning rate scheduling with 5 epochs of linear warmup. For data augmentation, we apply AutoAugment and CutMix to improve generalization.

**Methodology.** We apply our NerVE framework to compute eigenspectrum metrics (pre- and post-activation) for all 12 channel-mixing MLPs (dimension 3072) at epochs 1, 10, 20, 40, 80, and 120. We use full-batch covariance estimation with no sampling. Hence, at each logging epoch, *we accumulate the statistics across the entire training dataset* in a single forward pass which yields covariance matrices from 3.2M samples per layer. Our GPU-optimized implementation accumulates statistics for all layers simultaneously, requiring $\sim$432MB of additional GPU memory.

For an extensive eigenspectral dynamics study, we evaluate four architectural variants by systematically varying the activation functions in the token-mixing (FFN1) and channel-mixing (FFN2) layers: (1) GELU, GELU (baseline); (2) GELU, ReLU; (3) ReLU, GELU; and (4) ReLU, ReLU. We apply the full NerVE analysis to all configurations, tracking eigenspectrum metrics across all 12 layers. Figure 22 shows the eigenspectrum dynamics for each configuration, demonstrating how activation function choice impacts spectral properties during training.

**Observations.** First, across all four settings, post-activation eigenspectrum consistently exhibit higher spectral entropy and participation ratio than their pre-activation counterparts, with the gap increasing rapidly in the first 20 to 40 epochs, indicating that the Mixer nonlinearities reliably expand the effective dimensionality of the FFN representations rather than merely reshuffling.

Second, swapping GELU for ReLU in the channel-mixing FFN (FFN2) has a much stronger effect than changing the token-mixing FFN (MLP1). Configurations with ReLU in FFN2 show larger post-activation SE/PR and a more significant drop in EEE, indicating a flatter, less top-heavy spectrum where variance is redistributed away from the leading eigenmodes.

Third, the GELU (in FFN1) and ReLU (FFN2) variant, 3rd row in Figure 22, results in most aggressively flattened spectra indicated by highest post-activation PR and lowest post-activation EEE values while maintaining small JS divergence across most layers, suggesting that ReLU in token-mixing FFN drives a more uniformly expanded, higher effective dimensionality latent space rather than inducing sharp layer-specific distortions.

Finally, JS divergence is concentrated mostly in early layers when feature-mixing (FFN2) as GELU nonlinearity, whereas ReLU in FFN2 has more uniform pattern for layerwise JS divergence, mostly concentrated in deeper layers. This suggest that activation choice mainly modulates how much the boundary layers reshape the eigenspectrum while the interior layers act as relatively stationary propagators of the learned latent representation.

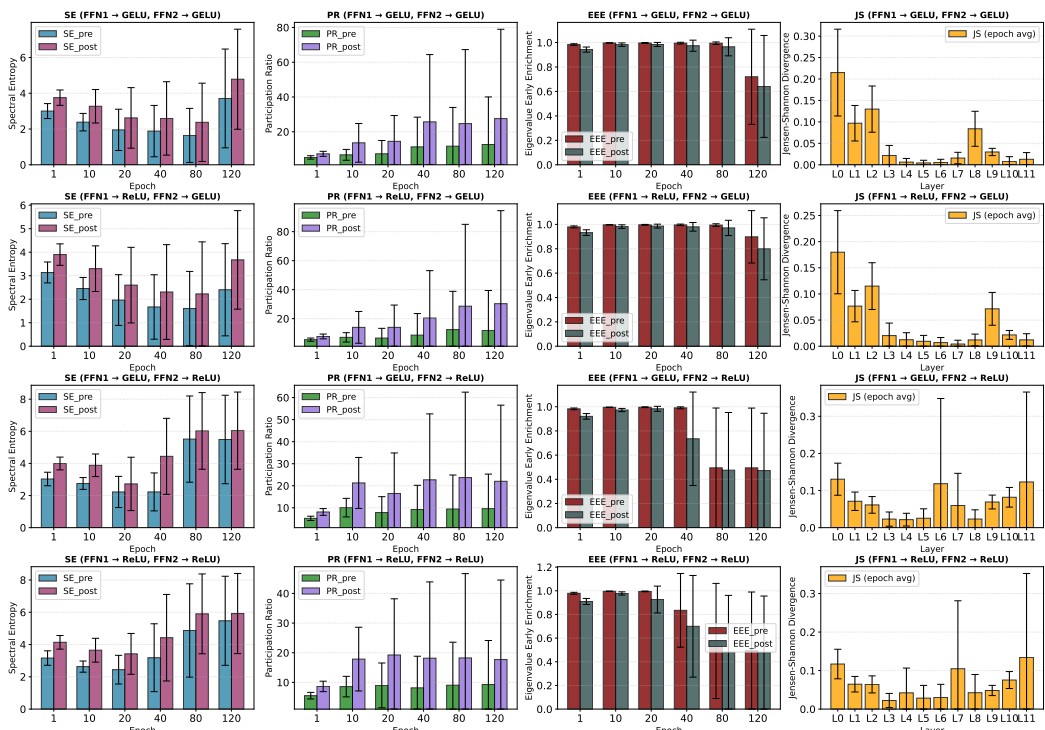

Figure 22: **Eigenspectrum dynamics in MLP-Mixer under activation ablations.** Rows correspond to the four activation configurations for token-mixing (FFN1) and channel-mixing (FFN2) layers, and columns (from left to right) show spectral entropy (SE), participation ratio (PR), eigenvalue early enrichment (EEE), and Jensen-Shannon divergence (JS) for the channel-mixing FFNs (FFN2). Each panel traces pre- and post-activation metrics over training, showing that ReLU in the channel-mixing MLP (3rd and 4th rows) most strongly increases SE/PR and reduces EEE, i.e., reinjects variance into low-energy directions and flattens the spectrum.

**Accuracy on CIFAR-100** Table 12 shows the accuracy for activation ablation study in MLP-Mixer. The ReLU in FFN2 leads to a better accuracy since it flattens the spectrum (lower EEE_post) and reduce the top-heaviness of the spectrum. Whereas, GELU in FFN2 leads to a higher EEE values throughout the training. The combination of GELU in FFN1, and ReLU in FFN2 leads to highest PR and lowest EEE, indicating best utilization of latent space in FFN2 and results in best accuracy.

Table 12: Validation accuracy on CIFAR-100 for different activation function configurations in MLP-Mixer. FFN1 refers to token-mixing MLP, FFN2 refers to channel-mixing MLP.

| FFN1 | FFN2 | Acc. (%) |
|------|------|----------|
| GELU | GELU | 66.96 |
| GELU | ReLU | 66.99 |
| ReLU | GELU | 67.86 |
| ReLU | ReLU | 67.20 |

# J  TOKEN-POSITION EFFECTS ON FFN EIGENSPECTRUM

To make the FFN eigenspectrum analysis explicitly sensitive to sequential structure, we stratify tokens by their position and recompute NerVE metrics within each position groups during their training. For GPT-2, we partition tokens into three groups along the sequence dimension (early, middle, late). For MLP-Mixer on CIFAR-100 we instead split patches into top vs. bottom rows of the $4\times4$ patch grid. For each configuration we then track effective dimensionality of the pre- and post-eigenspectrum using participation ratio metric, both aggregated across layers and averaged per layer (see Figure 23).

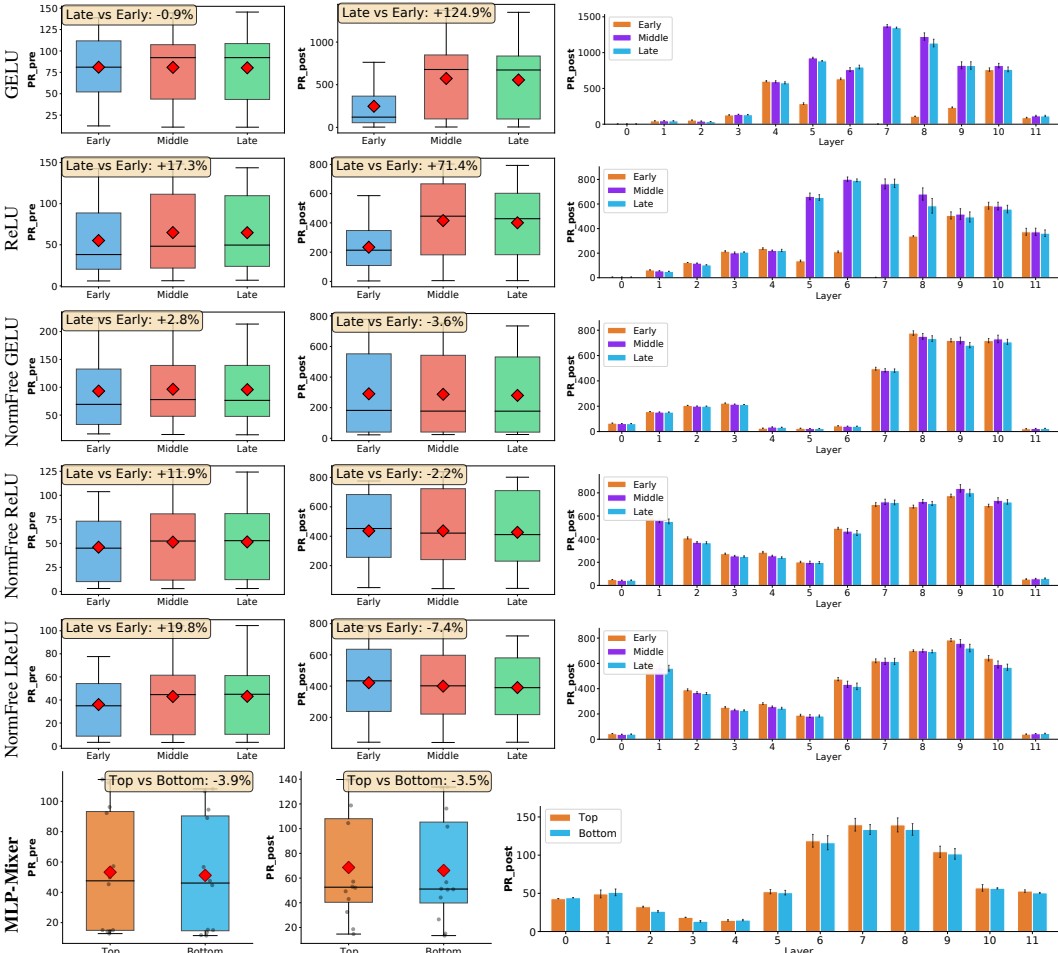

Figure 23: **Sequence-aware FFN eigenspectrum across token positions.** For four GPT-2 (125M) variants (rows 1 to 4) and a non-transformer MLP-Mixer (row 5), tokens are grouped by position. **Left**: pre-activation $\mathrm{PR}_{\mathrm{pre}}$; **Middle**: post-activation $\mathrm{PR}_{\mathrm{post}}$; and **Right**: layer-wise $\mathrm{PR}_{\mathrm{post}}$. In GPT-2 with baseline, both GELU and ReLU variants, early/middle/late tokens have similar $\mathrm{PR}_{\mathrm{pre}}$ but **diverge strongly** in $\mathrm{PR}_{\mathrm{post}}$, with late tokens using substantially higher effective dimensionality, mainly visible in deeper layers. However, in the normalization-free GPT-2 variants, these position-conditioned differences are largely suppressed in both pre- and post-activations. On the other hand, in a non-transformer family model, MLP-Mixer trained with SGD optimizer, where we split patches into top vs bottom rows, position effects are **weak** throughout. *This show that strong position-dependent FFN geometry is a characteristic of LayerNorm-based GPT-2*

**Baseline GPT-2 exhibits strong position dependence only after the FFN nonlinearity.** In both GPT-2 baseline variants (GELU and ReLU), the pre-activation PR shows almost no systematic differences between early, middle, and late tokens: the three boxes largely overlap, suggesting that the FFN representations before nonlinearity have comparable effective dimensionality across positions. In contrast, the post-activation PR shows a distinctive ordering: late tokens consistently exhibits a substantially higher PR than early tokens, with middle tokens in between. This effect is particularly pronounced for GELU, where late-token PR_post is roughly twice that of early tokens, and remains

visible under ReLU. The layerwise trend (right column) further reveal that this distinction emerges mainly in the second half of the network: *deeper FFNs layers allocate more latent degrees of freedom to later tokens, while early layers treat positions more uniformly.*

**Normalization-free GPT-2 suppresses position-dependent FFN geometry.** When we remove LayerNorm layers (NormFree-GELU and NormFree-ReLU), the position-induced differences reduce dramatically. Both pre- and post-activation PR distributions for early, middle, and late tokens nearly collapse onto each other, and the layerwise trends show very small gaps between position groups at all depths. That is, once normalization is removed, the FFN eigenspectrum become almost position-agnostic. This contrast suggests that in standard GPT-2, LayerNorm coupled with the nonlinearity amplifies positional biases in the FFN latent space, which is largely disappears in the normalization-free settings.

**Non-transformer MLP-Mixer shows only weak spatial position effects.** For the MLP-Mixer model, we perform an analogous top-bottom split over patch positions in baseline model (GELU in both MLPs) when trained from scratch on CIFAR-100. Here, position-conditioned PR differences are small in both pre- and post-eigenspectrum, and the layerwise PR trend for top vs bottom patches almost coincide. This indicates that the strong position dependence observed in GPT-2 FFNs is *not a generic property of FFNs:* in a non-transformer architecture trained on images, NerVE sees only mild spatial variation in FFN eigenspectrum.

# K    FFN Eigenspectrum Dynamics: LayerNorm vs RMSNorm

## K.1    LayerNorm vs RMSNorm Ablation on GPT-2

To examine how sensitive our FFN eigenspectrum analysis is to the normalization choices, we systematically vary both the placement of the normalization (PreLN, PostLN, MixLN) and the FFN activation type (GELU, ReLU, learnable leaky ReLU), and compare their effective post-activation dimensionality using the (Figure 24).

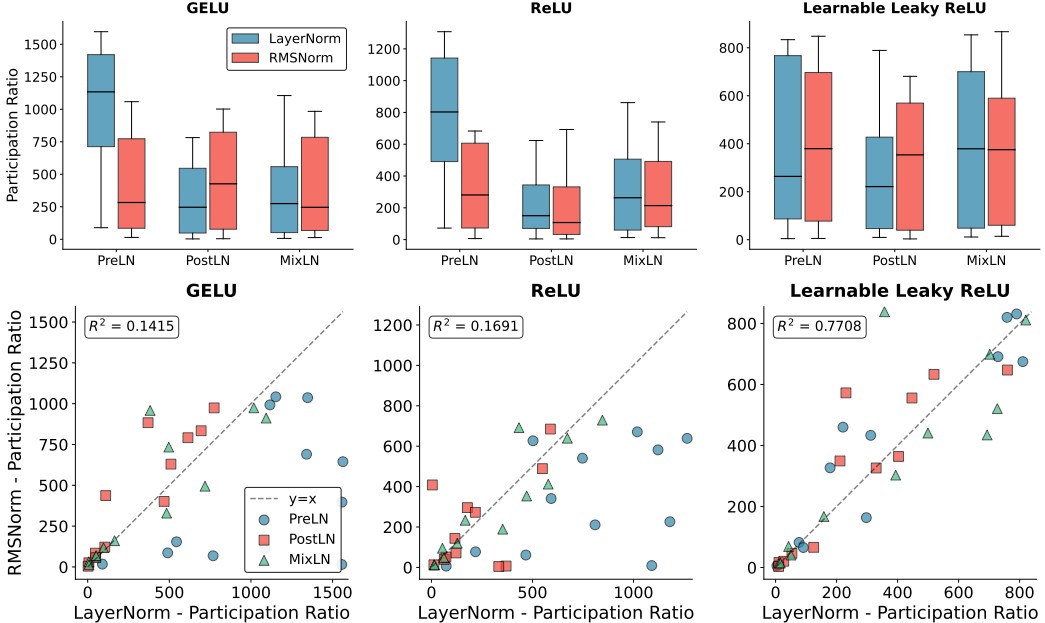

Figure 24: **Effect of LayerNorm vs RMSNorm on FFN eigenspectrum**. **Top row:** Post-activation participation ratio ($\text{PR}_{\text{post}}$) in GPT-2 (125M) for three FFN activations (GELU, ReLU, learnable leaky ReLU), three normalization placements (PreLN, PostLN, MixLN), and two normalization variants. Within each activation-placement pair, the LayerNorm and RMSNorm boxplots substantially overlap, while much larger shifts arise from changing the activation type or the norm placement. **Bottom row:** Layerwise comparison of $\text{PR}_{\text{post}}$ under LayerNorm ($x$-axis) vs RMSNorm ($y$-axis). Each point represents one FFN layer. GELU and ReLU show similar PR ranges but weak layerwise correspondence ($R^2 < 0.2$), while learnable leaky ReLU exhibits strong alignment ($R^2 \approx 0.77$). *This indicates that FFN eigenspectrum patterns are primarily driven by activation type and are largely robust to the choice of normalization scheme.*

**FFN activation type and the placement of normalization layers is more consequential than LayerNorm vs RMSNorm.** Figure 24 (top row) shows the distribution of post-activation participation ratio in various normalization layer positioning and FFN activation settings. Within each activation-placement pair, the LayerNorm and RMSNorm boxplots substantially overlap: their medians and interquartile ranges are of similar magnitude, and both exhibit the same depth-wise trend. For instance, PreLN exhibits highest PR spread for GELU while PostLN being more compressed.

By contrast, the *activation* choice induces much larger shifts in PR: GELU yields the largest PR values overall, ReLU compresses the spectrum, and learnable leaky ReLU lies in between with a narrower spread. Likewise, changing the PreLN to PostLN or MixLN has a clear effect on the PR distribution, whereas swapping LayerNorm to RMSNorm within a fixed placement produces only second-order changes, not the qualitative trends.

**Layerwise eigenspectral structure is largely preserved for LayerNorm vs RMSNorm.** Figure 24 (bottom row) compares LayerNorm and RMSNorm on a per-layer basis by plotting. For GELU and ReLU, the points cluster in a similar numeric range and follow a weak positive trend ($R^2 \approx 0.14$ and $R^2 \approx 0.17$, respectively): layers that are high-PR under LayerNorm tend to remain high-PR under RMSNorm, and low-PR layers remain low-PR, even though RMSNorm reshapes individual values somewhat. For learnable leaky ReLU, the alignment is much stronger ($R^2 \approx 0.77$). This shows

that the *ordering* of layers and the qualitative depth-wise behavior of the FFN eigenspectrum are (quantitatively)robust for LayerNorm vs RMSNorm; the main axes of variation remain activation type and the placement of normalization layer.

Recall that NerVE operates on *centered* FFN activations: when we construct (pre-/post-)covariance matrices, we subtract the empirical mean across tokens. This removes the DC component for eigenspectral analysis regardless of whether the model uses LayerNorm (which performs centering inside the model) or RMSNorm (which does not). Combined with the empirical evidence above, these ablations reassure that our conclusions about FFN eigenspectrum dynamics do not rely on a particular normalization layer; they hold for both LayerNorm and RMSNorm.

### K.2 LAYERNORM VS RMSNORM ABLATION ON MLP-MIXER

We extend our LayerNorm vs RMSNorm comparison to MLP-Mixer trained on CIFAR-100. Figure 25 shows post-activation SE, PR, and EEE throughout training, and JS the final-epoch JS across layers.

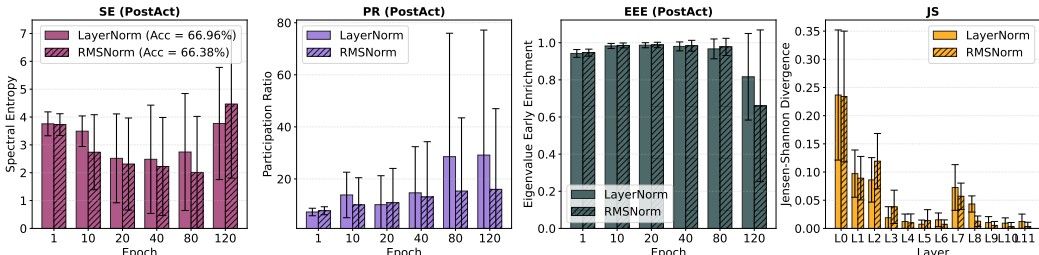

Figure 25: **Effect of LayerNorm vs RMSNorm in the MLP-Mixer models**. Columns (from left to right) shows post-activation SE, PR, EEE, and finally JS metric trend for LayerNorm and RMSNorm configurations throughout the training when MLP-Mixer models (GELU in both MLPs) are trained from scratch on CIFAR-100 dataset using Adam optimizer. FFN eigenspectrum dynamics remain (qualitatively) stable when LayerNorm is replaced with RMSNorm. However, in the later phases of training, the LayerNorm configuration attains higher effective dimensionality (PR) than RMSNorm, and its spectrum becomes noticeably flatter.

Throughout training, both normalization schemes produce very similar SE and EEE dynamics while layerwise JS trends at final convergence almost overlap, suggesting very similar reshaping of the eigenspectrum. The main difference appears in the final training phase, where the LayerNorm model attains a slightly higher PR_post and lower EEE_post than RMSNorm, and correspondingly achieves a modest accuracy gain (66.96% vs 66.38%). This suggests that, even in FFN-only architectures like MLP-Mixer, the qualitative eigenspectral behavior remain robust to the choice of normalization layer.

# L  OPTIMIZER-DEPENDENT FFN EIGENSPECTRUM DYNAMICS

## L.1  ADAMW VS MUON VS DION

We examine the optimizer-induced eigenspectral dynamics in GPT-2 160M models, when trained from scratch on FineWeb dataset with context size 512 (Figure 27) and 1024 (Figure 26) across AdamW, Muon (Liu et al., 2025; Shah et al., 2025; Boreiko et al.) and Dion optimizers, following the architectural and training hyperparameters settings from Ahn et al. (2025)

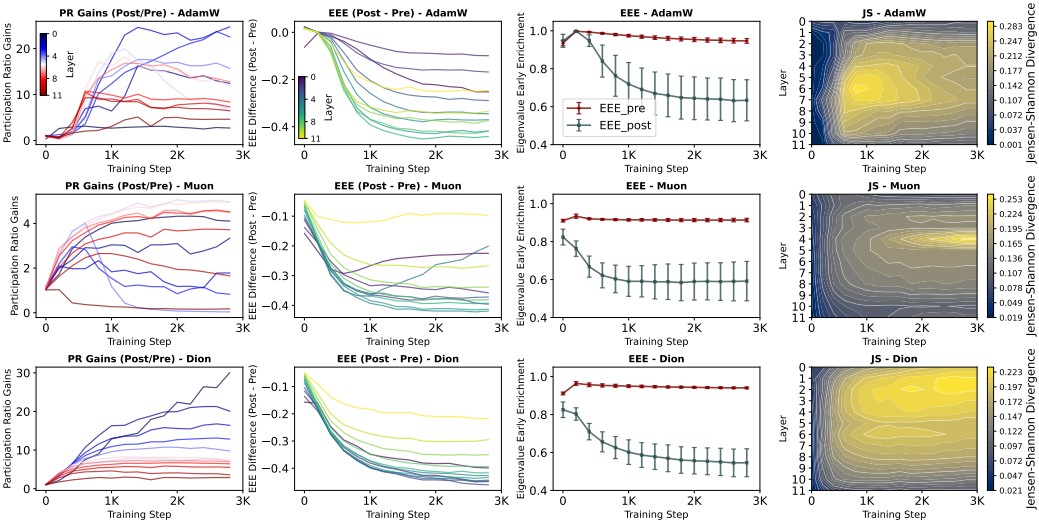

Figure 26: Optimizer-dependent FFN eigenspectrum dynamics in GPT2-160M (context length 1024). Rows show AdamW (top), Muon (middle), and Dion (bottom).

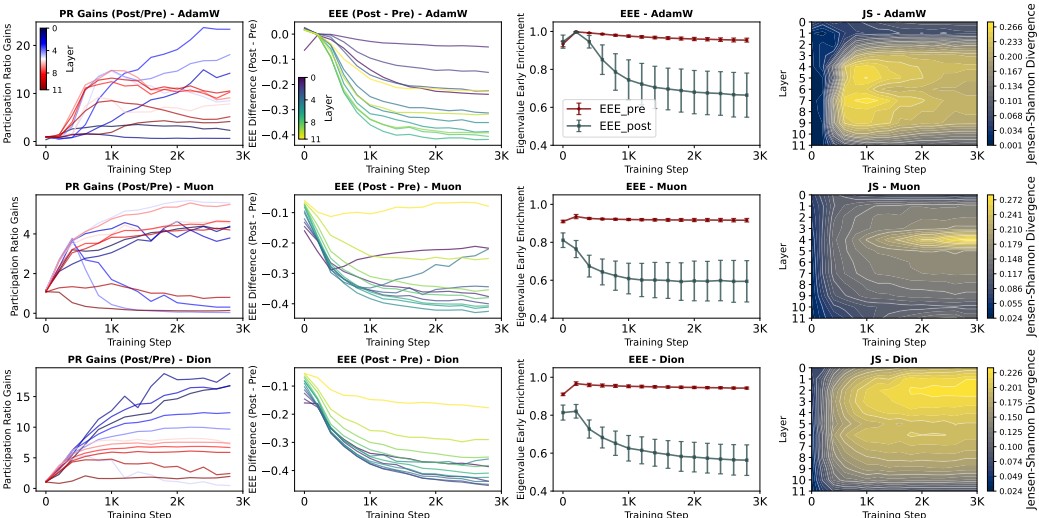

Figure 27: Optimizer-dependent FFN eigenspectrum dynamics in GPT2-160M (context length 512). Rows show AdamW (top), Muon (middle), and Dion (bottom).

Across context length 512 and 1024 in GPT-2 160M models, *a consistent eigenspectral trend emerges*. AdamW exhibits large early PR gains and high JS divergence with relatively high post-activation EEE (EEE_post), indicating optimizer-induced pre-activation collapse followed by aggressive but incomplete nonlinear repair. In contrast, Muon shows the smallest PR gains, lowest JS, and lowest EEE_post, with flatter post-activation spectra that suggest more stable eigenvalue distributions throughout training. Dion falls between these two regimes, it improves over AdamW but does not fully match Muon's spectral behavior. Notably, in a few early FFN layers, Dion exhibits PR gains on par with AdamW, suggesting layer-dependent repair dynamics.

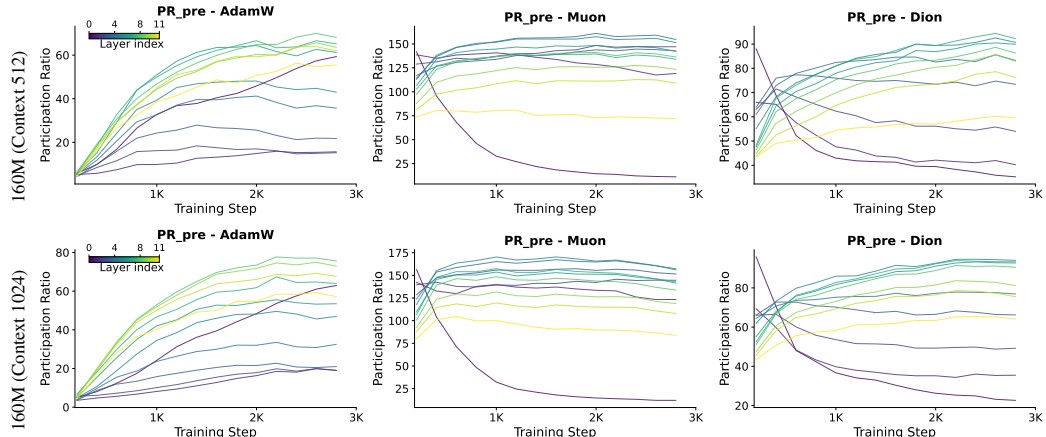

Figure 28: **Activation-compatibility of pre-activation eigenspectrum**. Columns show the effective dimensionality of pre-activation spectrum (PR_pre) when GPT2-160M models are trained from scratch with AdamW (left), Muon (middle), and Dion (right) optimizers, with 512 (top) and 1024 (bottom) context length. Muon exhibits the highest PR_pre across layers, demonstrating well-conditioned pre-activation spectra that place less burden on the FFN nonlinearity to restore representational rank.

To evaluate the conditioning of pre-activation eigenspectrum, and asses the extent to which FFN nonlinearity must actively inflate representational rank, we plot the layerwise evolution of PR_pre during training in Figure 28. Across settings, Muon consistently exhibits the highest effective dimensionality (PR_pre) across layers, indicating well-conditioned pre-activation spectra that place far less burden on the FFN nonlinearity to restore representational collapses. On contrary, AdamW shows pronounced early-layer collapse, while Dion partially mitigates these collapses.

Increasing the context length does not change this ordering; if anything, it makes the early-layer collapse under AdamW and Dion more pronounced, while Muon persistently keeps pre-spectra high-dimensional. This straighten the earlier observations that Muon produces activation-compatible representations across both model sizes and sequence lengths (Figure 8 and Figure 9).

Table 13: Evaluation perplexity (PPL) for GPT-2 models trained from scratch on the FineWeb dataset (Penedo et al., 2024) using AdamW, Muon, and Dion optimizer. Muon consistently achieves the lowest perplexity across all model sizes and context lengths, while AdamW results in worse perplexity.

| Optimizer | GPT2-350M (Context 512) | GPT2-160M (Context 512) | GPT2-160M (Context 1024) |
|---|---|---|---|
| AdamW (Loshchilov & Hutter, 2019) | 33.24 | 39.26 | 34.33 |
| Muon (Jordan et al., 2024) | 25.68 | 30.95 | 27.09 |
| Dion (Ahn et al., 2025) | 27.68 | 33.60 | 29.34 |

## L.2 ADAMW VS ADAFACTOR

To further investigate the optimizer-induced effects on FFN eigenspectrum, we substitute the AdamW with Adafactor (Shazeer & Stern, 2018) in GPT-2 and evaluate both the baseline and normalization-free variants with GELU and ReLU activations. We include Adafactor for optimizer-induced ablation study because designed to reduce optimizer memory via factorized second-moment estimates for matrix-shaped parameters, and offers a different preconditioning geometry from AdamW while remaining practical for training large language models (Raffel et al., 2020).

We begin with the baseline GPT-2 (125M) with GELU and ReLU activations to substantiate the core findings about the role of nonlinearity in FFNs (see Section 3.1). Figure 29 demonstrates the similar pre- and post-eigenspectrum characteristics that we observed earlier in Figure 1 and Figure 3. Precisely, the pre-activation spectrum entering the FFN are top-heavy and anisotropic, while the post-activation spectrum exhibit higher SE and PR, and a lower EEE, indicating that the FFNs systematically reinject variance and flatten the spectrum, counteracting the rank collapse induced by self-attention (Dong et al., 2021; Noci et al., 2022; Wu et al., 2024; Joseph et al., 2025).

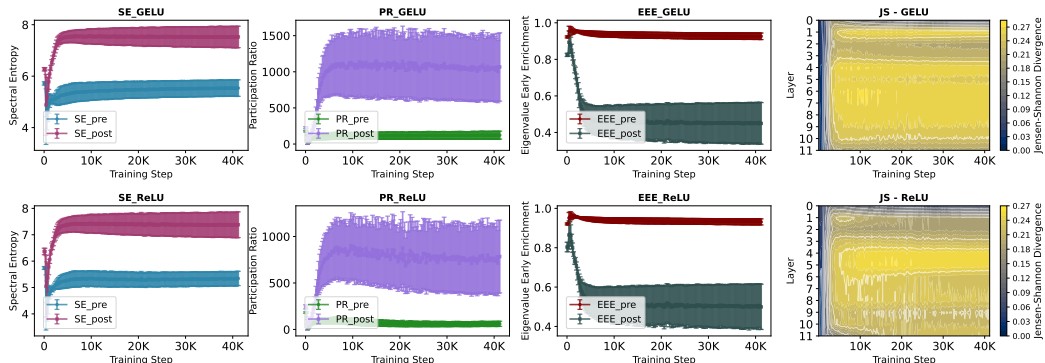

Figure 29: Eigen-metrics (SE, PR, EEE, and JS) illustrate *how FFN nonlinearities regulate information flow and reshape the eigenspectrum* when GPT-2 (125M) models with GELU (top) and ReLU (bottom) are trained from scratch on CodeParrot (2.1B tokens) datasets using **Adafactor** optimizer (Shazeer & Stern, 2018). Pre- and post-activation dynamics are shown for SE, PR, and EEE, highlighting how nonlinearities reinject variance and alter spectral structure. JS heatmaps (rightmost) capture the layerwise distributional shift induced by nonlinearity.

After establishing the fundamental role of FFN nonlinearity under both AdamW and Adafactor optimizer, we next compare the layerwise distribution of FFN capacity and latent-space utilization (PR_post) side-by-side in Figure 30 across four GPT-2 configurations (PreLN and Normalization-free, with GELU and ReLU activation). Notably, Adafactor consistently achieves higher post-activation PR than AdamW, and this gap is more pronounced in normalization-free ReLU model.

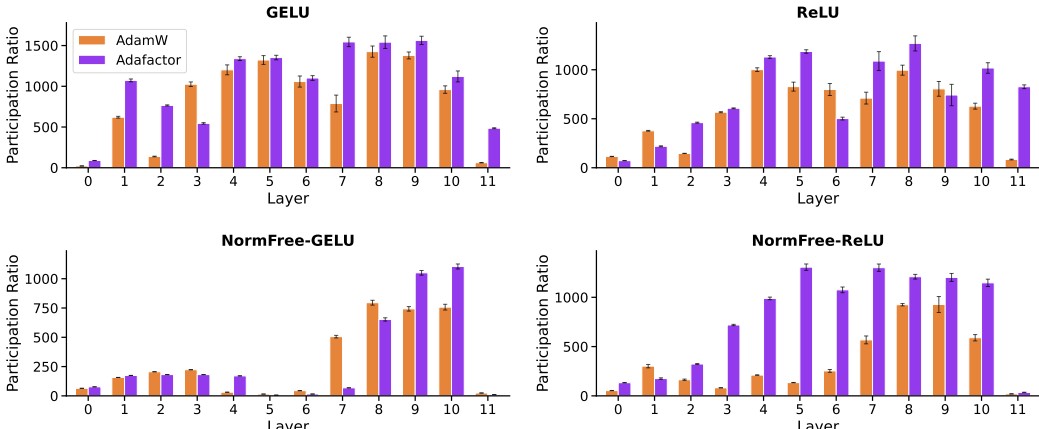

Figure 30: Layerwise post-activation participation ratio (PR_post) comparison for AdamW and Adafactor across four GPT-2 (125M) configurations: baseline (PreLN) with GELU and ReLU (top), and normalization-free GELU and ReLU (bottom). Across settings, Adafactor systematically produces higher PR_post than AdamW, indicating higher FFN latent-space utilization, with the largest gains in the normalization-free ReLU model.

This shows that, while FFN nonlinearity plays the same qualitative role—expands and flattens the spectrum—under both AdamW and Adafactor, the degree of this expansion is optimizer-dependent. Adafactor consistently drives stronger spectral expansion, activating more FFN latent capacity, especially in the normalization-free ReLU model.

Further, to contrast how effective dimensionality evolves under these two optimizers (AdamW and Adafactor), we plot the layerwise post-activation participation ratio (PR_post) over the entire training run (Figure 31). Across all four configurations, deeper layers quickly rise to a substantially higher PR value, and maintain that higher PR_post throughout the training. *This effect is consistently stronger with Adafactor than with AdamW*, indicating that Adafactor activates more latent capacity in the later FFNs throughout training.

We also track the change in EEE, $\Delta EEE = EEE_{post} - EEE_{pre}$, over training (Figure 32). More negative $\Delta EEE$ indicates stronger suppression of top eigenvalues and a flatter spectrum. Across all four

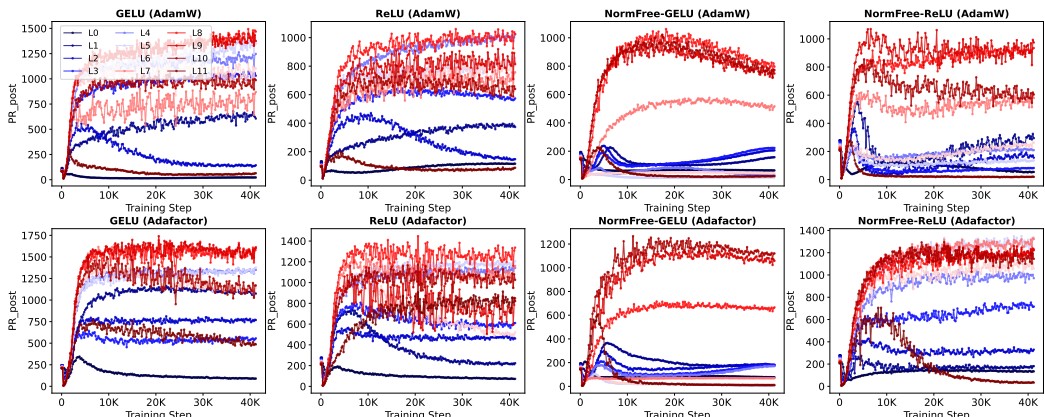

Figure 31: Training dynamics of post-activation participation ratio (PR) for all layers in GPT-2 (125M) across four configurations (columns) and two optimizers (rows). Deeper layers achieve and maintain higher PR_post throughout training, with Adafactor consistently producing larger PR_post than AdamW, especially in the deeper FFNs and in the normalization-free ReLU models.

GPT-2 configurations, both AdamW and Adafactor reduces ΔEEE well below zero, confirming the FFN's spectral flattening role. However, in the deeper layers, Adafactor consistently attains more negative ΔEEE than AdamW, especially in the normalization-free ReLU model, indicating a stronger reduction of top-heaviness and more aggressive spectral flattening in those FFNs.

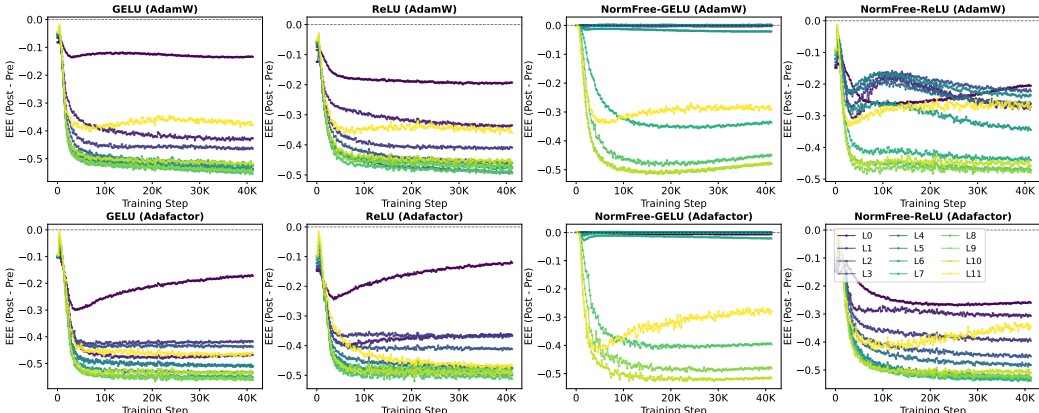

Figure 32: Training dynamics of FFN-induced spectral flattening in GPT-2 (125M), quantified as $\Delta EEE = EEE_{post} - EEE_{pre}$, under AdamW (top row) and Adafactor (bottom row) optimizer. More negative $\Delta EEE$ indicates stronger suppression of top eigenvalues and a flatter spectrum. Across settings, deeper layers under Adafactor tend to reach more negative $\Delta EEE$ than under AdamW, indicating more aggressive spectral flattening.

## L.3 ADAMW VS SGD

To examine how optimizers shape latent space utilization in a non-Transformer setting, we train the MLP-Mixer baseline (GELU in both MLPs) on CIFAR-100 with SGD and compare its eigenspectrum characteristics side-by-side with the Adam variant in Figure 33. For SGD, we use a learning rate of $5e\text{-}2$, momentum 0.9, and weight decay $1e\text{-}4$.

SGD outperforms Adam on MLP-Mixer (68.07% vs. 66.96%), and their spectral metrics trend explains this performance gap. Notably, SGD attains a significantly higher (post-activation) spectral entropy and participation ratio throughout training, indicating higher effective dimensionality and better utilization of FFN representational capacity. The EEE trend shows that Adam remains near 1.0 throughout training which suggests the persistent concentration of variance in the top eigenvalues. JS divergence further shows that Adam induces stronger spectral transformation in the very first layer, whereas SGD-induced transformation between pre- and post-eigenspectrum remain lower and more uniform across depth.

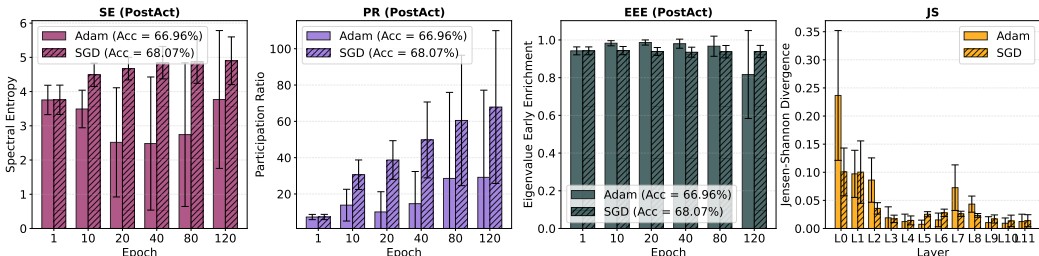

Figure 33: **Optimizer effects in MLP-Mixer (Adam vs. SGD)**. Columns (from left to right) shows post-activation SE, PR, EEE, and finally JS divergence metric trend when MLP-Mixer baseline model (GELU in both MLPs) is trained from scratch on CIFAR-100 dataset using Adam and SGD optimizer. Right from the epoch 10, the MLP-Mixer with the SGD optimizer starts exhibiting superior SE_post and PR_post showing significantly better utilization of FFN2 latent space.

## M    LIMITATIONS OF NERVE FRAMEWORK

**Per-layer independence.** Metrics are computed independently per layer, with no explicit measure of how spectra relate across depth. Our repair-vs-refinement findings (§3.7) suggest that optimizers can induce different degrees of cross-layer coherence, but NerVE does not currently quantify this. Cross-layer spectral-coherence measures (e.g., overlap between consecutive layers' top-$k$ subspaces) could formalize whether *smooth* spectral progression is associated with healthier training dynamics.

**Token-position aggregation.** By flattening the [B, S, D] tensor into a single [N, D] matrix, NerVE treats all token positions as exchangeable samples. This is a deliberate choice: covariance estimation benefits from the larger sample size, and the framework targets FFN-level geometry rather than position-specific dynamics. However, the position-stratified analysis (Appendix J) reveals that this aggregation masks substantial position-dependent structure in LayerNorm-based models (up to **+125%** PR difference between early and late tokens). When position-specific FFN geometry is the target, stratified analysis should complement the aggregate metrics.

**Practical constraints.** Full-batch covariance estimation and storing $D{\times}D$ matrices can be expensive at large scale. Our approximation study in Appendix G shows that token sub-sampling can preserve much of the *pre*-activation diagnostic signal, whereas low-rank truncation can distort correlation-based analyses, especially for *post*-activation metrics. For very large FFN dimensions (e.g., $D{>}10K$), practitioners should validate approximation fidelity for their specific setting and diagnostic objective.

## N    DISCUSSION: WHY TOP-HEAVINESS OVER TAIL-HEAVINESS

Heavy-tail (power-law) measurements are common in spectral analysis (Nair et al., 2022; Mahoney & Martin, 2019), but NerVE emphasizes *top-heaviness* because our goal is to characterize how FFN nonlinearities reshape the *dominant variance subspace* of activation covariances from pre- to post-activation. The leading eigenmodes capture directions that contribute most to second-moment energy and representation anisotropy. NerVE's EEE metric directly summarizes this dominant-subspace structure through the cumulative spectrum, providing a stable and interpretable measure of *front-loading* without requiring a tail cutoff.

In contrast, heavy-tail descriptors (e.g., fitting a power-law exponent) require choosing a tail range and performing goodness-of-fit checks. These choices can be sensitive and become harder to compare across scales. As FFN width $D$ varies, a fixed $k$ corresponds to different spectral fractions, and a fixed fraction corresponds to different absolute depths in the spectrum, complicating cross-configuration comparisons unless one performs careful per-scale calibration.

For these reasons, we do consider including tail-based measures in NerVE the framework. EEE is hyperparameter-free, numerically stable, and directly aligned with our central question: how nonlinearities transform dominant-subspace geometry and redistribute representational capacity across FFNs and training.

