# OpenReview forum: "NerVE: Nonlinear Eigenspectrum Dynamics in LLM Feed-Forward Networks"
_ICLR.cc/2026/Conference — ICLR 2026 Poster_

### Official Review · Reviewer_jJuL · 2025-10-19

**Soundness:** 3
**Presentation:** 2
**Contribution:** 2
**Rating:** 6
**Confidence:** 3

**Summary:**

This papers proposed to study the MLP/FFN sub-layers in LLM transformers using a spectral analysis. The authors argue that studying the activations immediately before and after the non-linearity (GeLU or ReLU or SwiGLU) allows one to capture the non-linear dynamics of the FFN. The authors proposed 4 metrics on the eigenspectrum of the covariance matrix that capture different effects (how top heavy the spectrum is, how much it has changed after the FFN). Studying these metrics in decode-only transformers shows several effects, most notably the FFN leads to more uniform spectrum/less top heavy eigenspectrum (e.g. fig 1). The authors also study several architectural design choices and how they affect these 4 metrics in some cases.

I am giving a 6 but would probably have given a 5 if there was the option.

**Strengths:**

- The topic of studying the effect of FFNs in LLMs is interesting and the idea of using the eigenspectrum dynamics to do so is well motivated.
- Several results have interesting insights such as in section 3.1, the FFN nonlinearity redistributes the variance across various eigenvalues.
- Studying several architectural choices such as layernorms and their positioning or spectral norm is interesting (although also see "weaknesses" about how thorough this is).

**Weaknesses:**

1. Overall, I think the paper identifies an interesting area but feels somewhat underdeveloped. Some particular areas where this comes across:

a. There doesn't seem to be a consistent message across section 3.2-3.5. Each of the subsections looks at a different architectural choice but the conclusions don't seem to tie together, e.g. how does the finding that rope prevents mid-to-deep spectral connect to the positioning of LN? For me the most interesting part of these sections was in section 3.4 that normalised participation ratio can be used as a diagnostic for healthy scaling, which I would maybe try to develop more into a consistent narrative across the section. For example, perhaps bad hyperparameter scaling choices (say the variance of the weight init or LR not scaling with width/depth) will become apparent via the normalised participation ratio not being consistent across scales, whereas with healthy scaling choices the PR is constant across scales (as seems to be with Pre-LN in figure 6)?
b. It feels like there could be some theoretical analysis here that could strengthen the foundations of the paper. E.g. what should one expect the dynamics of the Nerve metrics to be before/after a non-linearity under "healthy" behaviour and "unhealthy" behaviour. Should it be scale invariant?
c. I would study mixture of experts first before choosing to study e.g. RoPE or LN positioning, when discussing the effect of FFN architectural choices. This feels like an oversight. If it is too expensive to train and MoE then it should be possible to just take an open-source pretrained one and study the final checkpoint (or multiple checkpoints if available)?
d. Some of the claims are quite strong for being backed up by one experiment. E.g. in the discussion around hyperspherical normalisation the authors write "EEE post values remain high across depth, indicating the persistence of dominant directions despite the extended capacity", which doesn't seem obvious to me looking at Figure 5 (the early layers seem higher in weight and spectral normalisation).
e. I think the choices of Pre-FFN and Post-FFN are fine (at the top of page 3), but would also be interested to know what would happen if one studied x itself or FFN(x). Maybe more interesting for the latter is to study x + FFN(x) as maybe it's most important to understand how the FFN affects the residual stream?

2. It could be argued that spectral entropy, participation ratio, and early eigenvalue enrichment all look at a similar quantity (how dominant leading eigenvalues are). Indeed, in the plots (say figures 1 or 3) the first three columns look correlated. I'd ask what the benefit of these three separate metrics is, especially given that they add a lot of acronyms to the paper which the reader has to juggle around in his/her head, which can be confusing.

**Questions:**

1. What is the effect of RMSNorm not LayerNorm, as 3.2 is motivated by the centering of LN (which does not exist in RMSNorm).
2. Likewise, what if one uses a different optimiser than Adam?
3. "The EE score is... average vertical distance" is this not just the area between?

typos:
- "quantifies" not quantify line 201
- "yields" not yield line 222
- "t' in line 69

---

> ### Author Response · Authors · 2025-12-03
> **Authors' Offical Response**
>
> We thank the Reviewer **jJuL** for their time and a detailed feedback. Below, we address the key concerns.
>
> >What is the effect of RMSNorm not LayerNorm, as 3.2 is motivated by the centering of LN (which does not exist in RMSNorm).
>
> We have performed the RMSNorm vs LayerNorm experiments across **18 different settings in GPT-2** (PreLN/PostLN/MixLN and  GELU/ReLU/Learnable-LeakyReLU), and also on non-transformer family model MLP-Mixer. Results are presented and discussed in the Appendix J, and in **Global Response  "(4) Normalization variants (LayerNorm vs RMSNorm)"**.
>
> We find that FFN activation type and the placement of normalization layers matter more than the types of normalization layers (LayerNorm vs RMSNorm). In MLP-Mixer models, trained on CIFAR-100 dataset, layerwise eigenspectral structure is largely preserved for LayerNorm vs RMSNorm. This aligned with the observation made in the recent work (see Appendix G.4 in [1]).
>
>
> [1] Kim et al., Peri-LN: Revisiting Normalization Layer in the Transformer Architecture, ICML 2025
>
>
>  >Likewise, what if one uses a different optimiser than Adam?
>
> Thanks for asking this interesting question. We have done an extensive experimental study with different optimizer family: (a) AdamW vs Muon/Dion in GPT-2 350M and GPT2-160M, trained on FineWeb dataset; (b) AdamW vs  Adafactor in GPT-2 125M, trained on CodeParrot dataset; and (c) Adam vs SGD in MLP-Mixer models, trained on CIFAR-100 dataset.
>
>
> We have added the results in the main paper **Section 3.7** and **Appendix K**.  Please refer to **Global Response (5) Optimizer Robustness** for pointwise discussion on the key findings across different family of optimizer.
>
> In summary, these studies show that NerVE’s core findings about the role of FFN non-linearity---reinjecting the variance in the latent space and activates dormant directions; thus, increasing SE and PR, and flatting the spectrum by reducing the EEE post-activation---are optimizer-agnostic. However, different optimizers modulate how strongly FFN nonlinearities reshape the spectrum, which decides whether nonlinearities are in repair mode or refine model.
>
> **Note:** We did not include SGD in our LLM experiments, as prior work has shown that it performs worse in transformer-based LLMs due to strong block-wise curvature heterogeneity [2], and heavy-tailed class imbalance [3].
>
>
> >"The EEE score is... average vertical distance" is this not just the area between?
>
> Thank you for this observation and highlighting this geometric interpretation. You are correct that the mathematical computation is equivalent to the area between curves.
>
> However, we deliberately chose  the notion of *average vertical distance* for following reasons: (a) It emphasizes the discrete nature of eigenspectrum and eigenvalue computation. We measure deviation at D discrete eigenvalue indices rather than computing a continuous integral, which aligns with how eigenspectrum are computed in practice; and (b) For analyzing FFN eigenspectrum dynamics, the *per-eigenvalue deviation* interpretation provides clearer intuition, when post-activation EEE decreases, each eigenvalue position moves closer to uniform allocation, directly supporting our claims for variance redistribution.
>
>
>
> [1] Marbut et al., Reliable Measures of Spread in High Dimensional Latent Spaces, ICML 2023
>
>
> [2] Zhang et al., Why Transformers Need Adam: A Hessian Perspective, NeurIPS 2024
>
>
> [3] Kunstner et al., Heavy-Tailed Class Imbalance and Why Adam Outperforms Gradient Descent on Language Models, NeurIPS 2024

---

### Official Review · Reviewer_KsxW · 2025-10-30

**Soundness:** 3
**Presentation:** 3
**Contribution:** 3
**Rating:** 6
**Confidence:** 3

**Summary:**

The paper introduces "NerVE," a framework to analyze LLM feed forward networks by tracking the eigenspectrum of their pre- and post-activation covariance matrices. Using four metrics (spectral entropy, participation ratio, eigenvalue early enrichment, and JS divergence), it argues that the FFN's primary role is to "reinject variance" by taking the top-heavy output of attention and "flattening" its spectral distribution. This "spectral reshaping" increases the effective dimensionality for the next layer. The authors use this framework to provide geometric explanations for the effectiveness of architectural choices like Pre-LayerNorm and RoPE.

**Strengths:**

1. The paper's core claim—that FFNs function as spectral reshapers to re-awaken inactive dimensions is a compelling and intuitive explanation for their role. It provides a strong conceptual model that moves beyond viewing FFNs as simple key-value memories.
2. The chosen suite of four metrics is a strength. While SE and PR are related, the addition of EEE (to distinguish between different types of flat spectra) and, crucially, JS Divergence (to quantify the nonlinearity's effect) provides a more complete picture than any single metric.
3. The experiments effectively isolate variables. The norm-free analysis (Section 3.2), for instance, is a good way to demonstrate the FFN's compensatory role, showing that ReLU's piece-wise linear nature provides a regularization effect that the smoother GELU lacks.

**Weaknesses:**

1. The paper makes claims about "LLMs" but bases its findings on very small models (70M-130M). These spectral dynamics are not guaranteed to hold at the 1B+ parameter scales where architectural optimization is most critical. The findings need to be validated on larger models.
2. The paper repeatedly shows that "healthy" spectra (high PR, low EEE) correlate with low validation loss but fails to prove causation. It's just as likely that a well-optimized model produces these spectra as a byproduct of good performance. A direct intervention study (e.g., a spectral regularizer) is needed to make a causal claim.

**Questions:**

1. Seems the four metrics (entropy, participation ratio, eigenvalues, and JS divergence) are frequently used in related interpretability works. Can the author differentiate itself from other research that apply these similar metrics? Can the author provide a more detailed related works including what metrics are used for study what kind of phenomenon?
2. As mentioned in weakness 1, have the authors validated your key findings—particularly the efficiency of Pre-LN and the anti-collapse function of RoPE—on any models larger than 1B parameters?
3. As mentioned in weakness 2, have the authors considered an intervention study (e.g., adding a spectral regularizer) to demonstrate that enforcing a "healthy" spectral signature causes better model generalization, rather than just correlating with it?

---

> ### Author Response · Authors · 2025-12-03
> **Authors' Offical Response**
>
> We thank the reviewer **KsxW** for the positive evaluation of our work and their time. Below, we address the key concerns.
>
> >Can the author differentiate itself from other research that apply these similar metrics? Can the author provide a more detailed related works including what metrics are used for study what kind of phenomenon?
>
> While individual spectral metrics exist in various forms, our contribution is their unified application to FFN nonlinearity dynamics---a previously unexplored area---not the individual metrics. Here, we provide a brief discussion on the prior work.
>
> Poole et al. [1]  analyze nonlinear deep networks through a Riemannian/dynamical-systems lens, showing that the nonlinearity’s derivative controls an order–chaos transition and exponential growth of manifold curvature with depth.
>
> A recent work [2] tracks token-level cosine similarity and Lyapunov exponents for signal propagation, treating token evolution as a particle system to predict trainability from initialization.
>
>
> Marbut et al., [3] systematically evaluated the eight metrics---including eigenvalue ratio,  eigenvalue early enrichment (EEE), and other entropy and KL divergence based metrics--- to quantify the spread in high-dimensional  latent space on synthetic distributions. and they recommend one eigenvalue-based measure (EEE) as a reliable metric.
>
> Dong et al. [4] used relative residual norms and path decomposition to prove attention rank collapse, and showing the role of FFN in transformer-based model to prevent rank collapse.
>
> Zhang et al. [5] also used JS divergence to measure differences between Hessian spectrum of different parameter blocks in CNNs, GPT-2, ViT,  BERT, and MLP-Mixer, at initialization for predicting optimizer choice.
>
> >As mentioned in weakness 1, have the authors validated your key findings—particularly the efficiency of Pre-LN and the anti-collapse function of RoPE—on any models larger than 1B parameters?
>
> We have validated the efficient of PreLN on LLaMA-1.3B model, and contrast their spectral signature with MixLN counterpart. The spectral  signature is shown in **Figure 11**, and the results are discussed in **Appendix C.2**
>
> >As mentioned in weakness 2, have the authors considered an intervention study (e.g., adding a spectral regularizer) to demonstrate that enforcing a "healthy" spectral signature causes better model generalization, rather than just correlating with it?
>
> We appreciate the reviewer’s suggestion to go beyond analysis and explore explicit spectral interventions. In this work, we deliberately focused on establishing a clear, robust diagnostic picture of how FFN nonlinearities reshape the eigenspectrum across layers, optimizers, and architectures.
>
> While we did not implement direct spectral regularization, our experiments with spectral normalization (Section 3.3) provide partial evidence for this direction. Specifically, we found that spectral normalization, which regularize the spectral norms of weight matrix, in FFNs of LayerNorm-free GPT-2 model  achieved the best performance among other parametric normalizations by inducing **early and consistent spectral flattening** with lower EEE values and balanced PR.
>
> This suggests that architectural interventions targeting spectral properties can indeed improve generalization. However, a more direct approach would be adding a regularization term to  explicitly encourage flatter eigenspectrum  (lower EEE) and higher effective dimensionality (higher PR). For instance, the following:
>
> $$L_{spectral} = \lambda (EEE_{target} - EEE_{observed})^2 + \mu/PR_{observed}$$
>
> That said, designing and validating a generally applicable *spectral regularizer* that enforces a universally *healthy* signature  is **non-trivial for two reasons**: (a) our results show that the desirable PR/EEE profile is layer- and setting-dependent (e.g., mid-to-deeper layers and different optimizers exhibit systematically different spectrum), and (b) naively forcing the spectrum toward a fixed target curve may risks overfitting to our metrics rather than improving optimization and/or generalization.
>
> This is a promising future work, and we will add this to our discussion as an important next step for establishing causality. Thank you for this constructive suggestion,  it would indeed move beyond correlation to demonstrate that spectral signatures are not just indicators but actual drivers of performance.
>
> [1] Poole et al., Exponential expressivity in deep neural networks through transient chaos, NIPS 2016
>
> [2] Cowsik et al., Geometric Dynamics of Signal Propagation Predict Trainability of Transformers, Physical Review 2025
>
> [3] Marbut et al., Reliable Measures of Spread in High Dimensional Latent Spaces, ICML 2023
>
> [4] Dong et al., Attention is not all you need: pure attention loses rank doubly exponentially with depth, ICML 2021
>
> [5]  Zhang et al., Why Transformers Need Adam: A Hessian Perspective, NeurIPS 2024

---

### Official Review · Reviewer_yZ9g · 2025-11-01

**Soundness:** 2
**Presentation:** 3
**Contribution:** 3
**Rating:** 4
**Confidence:** 3

**Summary:**

- The paper introduces NerVE, a unified eigenspectral framework for analyzing feed-forward network (FFN) dynamics in LLMs.
- It tracks eigenspectrum dynamics of representations using Spectral Entropy, Participation Ratio, Eigenvalue Early Enrichment, and Jensen-Shannon Divergence to quantify variance dispersion, dimensionality, and nonlinear redistribution.
- The main finding is that FFN nonlinearities reinject and redistribute variance, activating underused dimensions and flattening eigenspectra to enhance latent space utilization, while demonstrating a correlation with validation loss.
- Experiments were run with GPT-2 and Llama-style architectures trained from scratch, showing that NerVE provides an interpretable, data-efficient tool for understanding FFN dynamics beyond empirical tuning.

**Strengths:**

- Provides a systematic, spectral lens to study FFN dynamics, an often-overlooked but important component of transformer models.
- The four complementary eigenspectrum metrics are theoretically interpretable, and capture distinct aspects of the representations
- The paper covers normalization variants, activation types, etc, making the analysis framework more broadly applicable.

**Weaknesses:**

- If I understand correctly, the paper treats activations from different sequences as interchangeable. In that case, what aspects of the analysis are specific to LLMs or transformer architectures? From this perspective, it may be valuable to also examine other types of models with FFNs and compare their behavior to that of LLMs. Alternatively, extending the analysis to explicitly account for the sequential structure of tokens could yield further insights.
- The models analyzed in the paper are relatively small by LLM standards (up to around 130M parameters). While it may not be feasible to train larger models from scratch, it could be informative to leverage open-weight models with available training checkpoints (e.g. Pythia) to study how the observed behaviors scale with model size.
- It would also be helpful to include some analysis on downstream tasks, such as computing the proposed metrics on datasets or domains unseen during training, to test whether the observed patterns generalize.
- Expanding the conclusion by summarizing the main findings and contributions would help consolidate the paper’s message.

**Questions:**

- When computing correlations with validation loss, are the spectral metrics measured on training data or separate held-out sets?
- Could a similar analysis be done on the other components of the transformer blocks? This may help link the inherent sequential structure of the tokens to your analysis
- Could NerVE’s metrics generalize to other architectures (eg. non-transformers such as ResNets)? This could test whether the observations are a general FFN property.
- Could you discuss some practical implications based on the proposed framework?

---

> ### Author Response · Authors · 2025-12-03
> **Authors' Offical Response**
>
> We thank the Reviewer **yZ9g** for a detailed readings of paper, and constructive suggestions and feedback. Below, we address the concerns raised during the review period.
>
> >When computing correlations with validation loss, are the spectral metrics measured on training data or separate held-out sets?
>
> Spectral metrics are always measured on training data, as co-variance matrices are created from sampling the pre-/post-activation during the training phase, and metrics computation is turned-off during the eval phase.
>
> >Could a similar analysis be done on the other components of the transformer blocks? This may help link the inherent sequential structure of the tokens to your analysis
>
> Yes, it can be done on any architectural block by registering the PyTorch hooks at appropriate places. For instance, NerVE can be used for understanding how FFN or Self-attention block transform the positional information, by logging the incoming and outgoing positional information to FFN/Self-attention block.
>
> To analyze the  effect of inherent sequential structure  of the tokens, we have performed an extensive study with the positional-specific information in eigenspectrum. Please refer to **Appendix I**, and **global response "(3) Sequence-aware (position-stratified) analysis"**
>
> >Could NerVE’s metrics generalize to other architectures (eg. non-transformers such as ResNets)? This could test whether the observations are a general FFN property.
>
> Indeed, NerVE's metric can be used on non-transformer architecture, and we have employed the NerVE's on channel-mixing FFN in the MLP-Mixer (B/16) models. We performed MLP-Mixer experiments in **6 different settings:** 4 configurations for activation (GELU/ReLU) in token-mixing and channel-mixing MLPs/FFNs, one for LayerNorm vs RMSNorms, and one for Adam vs SGD study.
>
> Results are discussed in Global response **"(2) Non-Transformer Architecture (MLP-Mixer)"** and **Appendix H, J.2, and K.3**
>
>
> >Could you discuss some practical implications based on the proposed framework?
>
> While NerVE is an analysis framework, it directly informs several concrete design and debugging choices for LLMs. First, NerVE provides a way to select and tune FFN nonlinearities and widths, by tracking how SE/PR/EEE/JS change from pre- to post-activation, one can detect over-compressive FFNs (excessively top-heavy eigenspectrum, low PR_post) versus under-utilized ones (minimal spectral change), and adjust activation type, hidden width, or LayerNorm placement accordingly.
>
>
>
> Second, our optimizer and normalization studies show that NerVE can be used as a diagnostic tool during training: patterns such as  pre-activation spectral collapse followed by aggressive nonlinear **repair** (large PR gains and EEE difference) flag unstable representation dynamics even when loss/perplexity curves look benign. This offers a complementary signal for choosing between optimizers (e.g., AdamW vs Muon/Dion) or normalization variants with well-conditioned  latent spectrum.
>
>
> Finally, because NerVE is architecture-agnostic (we demonstrate it on GPT-2, LLaMA, and MLP-Mixer), it can be used to evaluate architectural simplifications (e.g., pruning or linearizing FFNs) and to guide future methods that explicitly regularize eigenspectral properties, with the goal of preserving the representational capacity through better allocation of latent capacity along the network's depth, while reducing compute or nonlinear complexity.

---

### Author Response · Authors · 2025-11-21
**Reply to All the Reviewers**

Dear Reviewer **yZ9g, KsxW, jJuL,**


We are currently analyzing results from the new experiments, and a few runs are still in progress. We will update the draft with the new results, highlighting all the additions, and post it alongside the rebuttal no later than **Nov 26 (Wednesday) AoE**. Thank you for your understanding.

Best,

Authors.

---

### Author Response · Authors · 2025-12-02
**Global Response: New Experiments and Consolidated Takeaways (1/2)**

We thank the reviewer yZ9g, KsxW, and jJuL, for their detailed and constructive feedback. We begin by  restating the central theme of our paper: NerVE introduces a four-metric eigenspectrum framework (SE, PR, EEE, JS) to study how FFN nonlinearities reshape latent representations in LLMs.


Reviewers primarily asked whether our conclusions are robust across optimizer, normalization schemes, non-Transformer architectures, larger scale of LLMs (1B+), and sequence structure. During the rebuttal period, **we ran a series of targeted experiments** to address these points. This global response summarizes only these new results and their implications; the overall narrative and main claims of the paper remain unchanged but are now corroborated with additional  experiments.


We organize the new experiments below by theme:

(1) Larger scale of LLMs,

(2) Non-Transformer architecture,

(3) Sequence-aware (position-stratified) analysis,

(4) normalization variants, and

(5) Optimizer robustness

Below, we describe each experimental theme in a detailed pointwise manner, including the experimental setup and key takeaways.


## (1) Larger scale of LLMs (LLaMA-250M and LLaMA-1.3B)


We extend our NerVE analysis for PreLN and MixLN variants of **LLaMA-250M** and **LLaMA-1.3B** models, and results are described in **Appendix C.2**. Following the LLaMA architectural parameters and training settings from [1], we trained these LLaMA models from scratch on C4 dataset with 256 context length.


The spectral signatures for PreLN and MixLN variants of these larger LLaMA models are shown in **Figure 11**.  The  spectral utilization patterns explains why MixLN configuration results in lower perplexity for LLaMA-250 models, and also the training collapses at 1.3B scale.  Moreover, the heatmaps in Figure 11 also depicts layers suffering from representational bottleneck (very low spectral entropy).



## (2) Non-Transformer Architecture (MLP-Mixer)


To substantiate the generalizability of NerVE's findings (originally  employed only on transformer-based LLMs), we employ our NerVE framework on MLP-Mixer (B/16) models [2], and trained them from scratch on CIFAR-100 datasets **across six different configurations**.


**Appendix H** discusses our rationale for selecting the MLP-Mixer models as a non-transformer based architecture, training settings, and methodology for employing NerVE and collecting samples with full-precision. We would like to emphasize that for constructing the co-variance matrices (pre- and post-activation) **we forward pass the entire dataset (3.2M samples)**.

**Figure 21** demonstrate the eigenspectrum dynamics of channel-mixing MLP (FFN2) across 4 different settings for activation function (GELU and ReLU)   in the token-mixing and channel-mixing MLPs. We also describe our observations in Appendix H, and Table 7 shows the accuracy results on CIFAR-100.


The key observations details how the core claims made in the paper for the role of FFN nonlinearity persist even in non-transformer architecture, and how the spectral dynamics explains the predictive performance across settings.

## (3) Sequence-aware (position-stratified) analysis

This is one of the most-interesting experiential study performed (**Thanks to Reviewer yZ9g**) during the rebuttal period.  We perform (**Appendix I**) experiments on both GPT-2 and MLP-Mixer models to show the impact of token-positions on the key eigen metrics. For GPT-2 models we perform experiments also in normalization-free settings.


**Figure 22** shows the results, and Appendix I discussed the key findings: 1) Baseline GPT-2 exhibits strong position dependence only after the FFN nonlinearity (not in per-activation spectrum),  2) Normalization-free GPT-2 suppresses position-dependent FFN geometry, and 3) Non-transformer MLP-Mixer shows only weak spatial position effects. This shows that **position-dependent effect of tokens are not universal** (even in sequence-based transformer models).


## (4) Normalization variants (LayerNorm vs RMSNorm)


In **Appendix J** we describe the impact on substituting LayerNorms with RMSNorm in GPT-2 models across various settings: 1) Normalization layer's positioning (PreLN, MixLN, and PostLN), and 2) FFN activation (GELU, ReLU, and Learnable Leaky ReLU).


**Figure 23** shows the results for all the **18 different configurations** in GPT-2 models. The key takeaways from this experimental study is:  *FFN activation type and the placement of normalization layers matter more than the types of normalization layers (LayerNorm vs RMSNorm)*. This finding aligned with the observation made in the recent work (see Appendix G.4 in [3]).


[1] Li et al., Mix-LN: Unleashing the Power of Deeper Layers by Combining Pre-LN and Post-LN, ICLR 2025

[2] Tolstikhin et al., MLP-Mixer: An all-MLP Architecture for Vision, NeurIPS 2021

[3] Kim et al., Peri-LN: Revisiting Normalization Layer in the Transformer Architecture, ICML 2025

---

> ### Author Response · Authors · 2025-12-02
> **Global Response: New Experiments and Consolidated Takeaways (2/2)**
>
> We also perform the LayerNorm vs RMSNorm experiments with MLP-Mixer models and the results are shown in **Figure 24** **(Appendix J.2)**, and we find that even in FFN-only architectures like MLP-Mixer, the qualitative eigenspectral behavior remain robust to the choice of normalization layer.
>
>
> ## (5) Optimizer Robustness
>
>
> To understand how optimizer choice affects FFN eigenspectrum dynamics and the role of FFN nonlinearity, and to investigate whether the NerVE's findings depend on a particular optimizer, we conducted three sets of additional experiments:
>
> (a) AdamW vs Muon vs Dion on GPT-2 (350M and 160M) trained on FineWeb,
>
> (b) AdamW vs Adafactor on GPT-2 (125M) trained on CodeParrot, and
>
> (c) Adam vs SGD on MLP-Mixer trained on CIFAR-100.
>
> Across all settings, the qualitative role of the FFN nonlinearity remained same: it flattens the eigenspectrum and reduce top-heaviness (EEE_post < EEE_pre), and increases effective dimensionality (PR_post > PR_pre) and spectral entropy (SE_post >  SE_pre).  **However, the degree of these effects varies with the optimizer choices**. Below, we summarize the key findings from each of these three different set of  optimizer family.
>
>
>
> ### 5.a  AdamW vs Muon vs Dion
>
> The first set of experiments, for understanding the optimizer-induced eigenspectral behavior, consist of three optimizers: the standard AdamW baseline [4], the geometric optimizer Muon [5], and the low-rank geometric optimizer Dion [6].
>
> We train three architecturally identical **GPT2-350M** models (24L, 32H, 1024d) on FineWeb dataset, following the training settings from [6];  particularly,   AdamW (lr = 3e-3), Muon (lr = 2e-2), and Dion (lr = 2e-2, rank = 0.125). Also, we repeat the experiments on a smaller GPT2-160M model with context lengths 512 and 1024 to further verify the robustness.
>
> We have added the results for this set of experiments in main paper **Section 3.7 (Figures 8 and 9)**,  and **Appendix K.1 (Figures 25, 26, and 27; and Table 8)**.  These results show that optimizer geometry does not just change convergence speed; it changes how FFN nonlinearities are used.
>
> AdamW pushes FFNs into a repair regime, nonlinearity is spent undoing optimizer-induced spectral collapse, whereas Muon moves FFNs into a refinement regime, preserving activation-compatible pre-activation eigenspectrum and concentrating high-dimensional representations in the middle layers where model performance is most sensitive. Dion's spectral behavior lie between these two regimes, where the role of FFN non-linearity (repair vs refinement) is depth-dependent.
>
>
> ### 5.b  AdamW vs Adafactor
>
> We repeat the experiments, that originally performed with AdamW optimizer  in Section 3.1 and 3.2, with the Adafactor optimizer [9] for GPT-2 baseline and normalization-free models with GELU and ReLU activation.
>
> The experimental results are presented in **Appendix K.2 (Figures 28-31)**. We find that across all four configurations the qualitative NerVE story is identical under AdamW and Adafactor:  flattening the eigenspectrum and reduce top-heaviness, and increases effective dimensionality and spectral entropy.
>
> The main quantitative difference is that Adafactor systematically amplifies this effect: deeper FFNs trained with Adafactor attain higher PR\_post and more negative ΔEEE than with AdamW, especially in the normalization-free ReLU model, indicating better utilization of latent capacity and more aggressive suppression of top-heaviness.
>
> ### 5.c  Adam vs SGD
>
> The MLP-Mixer Adam [7] vs SGD is performed on CIFAR-100, and the results are discussed in Appendix **K.3 (Figure 32)**. SGD yields better accuracy than Adam, and their spectral dynamics demonstrate a better utilization of latent capacity (higher SE_post and PR_post) throughout the training.
>
>
> **In summary**, these studies show that NerVE’s core findings are optimizer-agnostic. Different optimizers modulate how strongly FFN nonlinearities reshape the spectrum, but not whether they perform rank-repair and spectral flattening.
>
> Note that prior work [10] showed that self-attention has a strong inductive bias towards token uniformity, while nonlinear MLP/FFN blocks partially undo this effect by slowing or reversing rank collapse. Our NerVE framework provides a complementary spectral view (rank-repair) focused on the FFN latent space.
>
>
>
> [4] Loshchilov et al., Decoupled Weight Decay Regularization, ICLR 2019
>
> [5] Jordan et al, Muon: An optimizer for hidden layers in neural networks, 2024
>
> [6] Ahn et al., Dion: Distributed Orthonormalized Updates, 2025
>
> [7] Kingma et al., Adam: A method for stochastic optimization, ICLR 2015
>
> [9] Shazeer et al., Adafactor: Adaptive learning rates with sublinear memory cost, ICML 2018
>
> [10] Dong et al., Attention is not all you need: pure attention loses rank doubly exponentially with depth, ICML 2021

---

### Meta-Review · Area_Chair_9tZT · 2026-01-02

**Summary:**

The paper introduces NerVE, a unified (empirical) framework for analyzing the nonlinear dynamics and role of feed-forward networks (FFNs) in LLMs.
It proposes four metrics to quantitatively characterize the eigenspectra (of representations) and provides a range of insights into the role of FFNs and nonlinearities in LLMs.

All three reviewers agree that the paper presents a systematic and well-motivated analysis framework and offers interesting and novel insights for LLMs.
While the reviewers raised several concerns (summarized below), the authors have done a good job and addressed most of them satisfactorily during the rebuttal.

**Reviewer Concerns:**

- Reviewer **yZ9g** asked for: (1) extension to non-transformer-based models, (2) larger LLMs, and (3) impact on (generalization) performance of downstream tasks; I believe that the authors have fully addressed the first two, and partially the third (which, I would say, does not hurt the contribution of this paper).
- Reviewer **KsxW** asked for: (1) larger LLMs and (2) impact on (generalization) performance of downstream tasks, or the explicit use of a spectral regularizer; I believe that the authors have fully addressed the first and partially the second (which, I would say, does not hurt the contribution of this paper).
- Reviewer **jJuL** asked for many things (different optimization schemes, theoretical analysis, MoE results, other more different metrics, etc.); I feel that some of these were addressed during the rebuttal, while others were not.

**Reviewer Scores:**

- I believe that reviewer **yZ9g** would have increased the score, as most of the concerns have been convincingly addressed.
- I believe that reviewer **KsxW** would have slightly increased the score or kept it unchanged (6, which is already positive).
- I believe that reviewer **jJuL** would have kept the score unchanged (6, which is already positive).

---

### Decision · Program_Chairs · 2026-01-26

Accept (Poster)